# Typicalness-Aware Learning for Failure Detection

**Yijun Liu**[1]    **Jiequan Cui**[2]    **Zhuotao Tian**[1✉]    **Senqiao Yang**[3]
**Qingdong He**[4]    **Xiaoling Wang**[1]    **Jingyong Su**[1✉]
{liuyijun}@stu.hit.edu.cn
[1]Harbin Institute of Technology (Shenzhen)    [2]Nanyang Technological University
[3]The Chinese University of Hong Kong    [4]Tencent Youtu Lab

## Abstract

Deep neural networks (DNNs) often suffer from the overconfidence issue, where incorrect predictions are made with high confidence scores, hindering the applications in critical systems. In this paper, we propose a novel approach called Typicalness-Aware Learning (TAL) to address this issue and improve failure detection performance. We observe that, with the cross-entropy loss, model predictions are optimized to align with the corresponding labels via increasing logit magnitude or refining logit direction. However, regarding atypical samples, the image content and their labels may exhibit disparities. This discrepancy can lead to overfitting on atypical samples, ultimately resulting in the overconfidence issue that we aim to address. To tackle the problem, we have devised a metric that quantifies the typicalness of each sample, enabling the dynamic adjustment of the logit magnitude during the training process. By allowing atypical samples to be adequately fitted while preserving reliable logit direction, the problem of overconfidence can be mitigated. TAL has been extensively evaluated on benchmark datasets, and the results demonstrate its superiority over existing failure detection methods. Specifically, TAL achieves a more than 5% improvement on CIFAR100 in terms of the Area Under the Risk-Coverage Curve (AURC) compared to the state-of-the-art. Code is available at https://github.com/liuyijungoon/TAL.

## 1   Introduction

Failure detection plays a vital role in machine learning applications, particularly in high-risk domains where the reliability and trustworthiness of predictions are crucial. Applications such as medical diagnosis [8], autonomous driving [19, 42, 43], and other visual perception tasks [22, 38, 32, 25, 35, 30, 37, 29, 36] require accurate assessments of prediction confidence before making critical decisions. The goal of failure detection is to enhance the reliability and trustworthiness of predictions, ensuring that high-confidence predictions are relied upon while low-confidence predictions are appropriately rejected [26, 47]. This is essential for maintaining the safety and effectiveness of these applications.

Indeed, deep neural networks (DNNs) trained using the cross-entropy loss often suffer from the issue of overconfidence. This leads to unreliable confidence scores, which in turn hinder the effectiveness of failure detection methods. It is common for models to make incorrect predictions with high confidence scores, sometimes even close to 1.0. A recent study called LogitNorm [40] has shed light on this problem. It reveals that the softmax cross-entropy loss can cause the magnitude of the logit vector to continue increasing, even when most training examples are correctly classified. This phenomenon contributes to the model's overconfidence.

To alleviate the overconfidence problem, LogitNorm [40] proposes assigning a constant magnitude to decouple the influence of the magnitude during optimization. The cross-entropy loss with the

---

✉: Corresponding Authors. Email: tianzhuotao@hit.edu.cn, sujingyong@hit.edu.cn

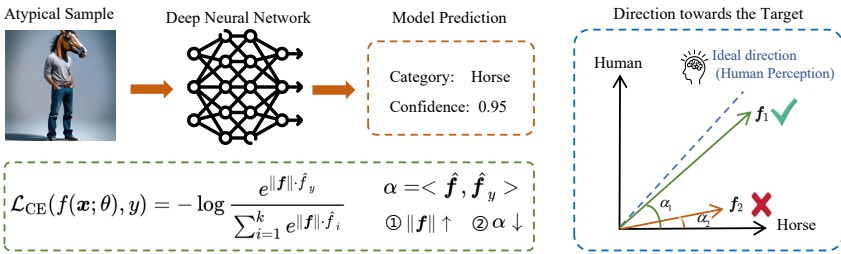

Figure 1: Illustration of the motivation. We observe that directly aligning the predictions of atypical samples to the target label is not appropriate, causing overconfidence (horse with 95% confidence). Instead, the confidence should be aligned with the human perception. During training, the cross-entropy loss increases the magnitude $\|\boldsymbol{f}\|$ and adjusts their direction towards the target (represented by the angle $\alpha$). Consider this example where an image of a human body with a horse head is presented, the loss may optimize towards $\boldsymbol{f}_2$ in the blue box, which is not the ideal outcome direction. Instead, it would be better to optimize towards $\boldsymbol{f}_1$, rather than being biased towards either one, ensuring a more balanced and unbiased representation and allowing for a more accurate estimation of confidence.

decoupled logit vector $\boldsymbol{f} = f(\boldsymbol{x};\Theta)$ is defined as what follows:

$$\mathcal{L}_{\text{CE}}(\boldsymbol{f}, y) = -\log \frac{e^{\|\boldsymbol{f}\| \cdot \hat{\boldsymbol{f}}_y}}{\sum_{i=1}^{C} e^{\|f\| \cdot \hat{\boldsymbol{f}}_i}}, \tag{1}$$

where $\Theta$ is the parameters of the DNN model, $\boldsymbol{x}$ is the input image with label $y$, the logit vector $\boldsymbol{f}$ is decoupled into the *magnitude* $\|\boldsymbol{f}\|$ and the *directions* $\hat{\boldsymbol{f}}$. Based on the decomposition of the logit vector in Eq. (1), we can observe that the overconfidence issue could stem from either increasing $\|\boldsymbol{f}\|$ or decreasing $\alpha$ (the angle between the directions of prediction and label) during training.

**Key observation & Motivation.** Following the exclusion of the logit magnitude's impact by LogitNorm [40], we posit that the risk of the overconfidence issue still arises from logit directions. Typical samples, which have clear contextual information, help models generalize well. However, optimizing the direction for ambiguous atypical samples can still cause overconfidence. In these cases, the labels do not match the image context well. Aligning the logit direction of atypical samples may still lead to high softmax scores near 1.0 which worsens the overconfidence problem.

According to previous work [51, 44], the definitions of typical and atypical samples are based on their semantic similarity to most samples and the ease with which the model learns them. Specifically:

- *Typical samples* are those that exhibit similarity to a majority of other samples at the semantic level. These samples possess typical features that are easier for deep neural networks to learn and generalize.

- *Atypical samples*, on the other hand, differ significantly from other samples at the semantic level. They pose a challenge for the model to generalize due to their uniqueness. These samples are often located near the decision boundary, causing the model to have higher uncertainty in making predictions for them.

In Fig. 1, we present an atypical example to illustrate the issue at hand. Despite the ground-truth label being a horse, the image depicts a horse with a human body, which could reasonably be predicted as either a human or a horse with a confidence score of around 50%. However, the model incorrectly predicts the image as a horse with an excessively high confidence score of 95%. Upon examining Eq. (1), we observe that the confidence score is determined by two crucial factors: magnitude and direction. This prompts an important question regarding the decoupling of these factors to determine which one is more reliable in accurately approximating real confidence. Addressing this inquiry is essential for effective failure detection.

**Our approach.** Based on the aforementioned observations, we propose a novel approach called *Typicalness-Aware Learning (TAL)*. TAL dynamically adjusts the magnitudes of logits based on the typicalness of the samples, allowing for differentiated treatments of typical and atypical samples. By doing so, TAL aims to mitigate the adverse effects caused by atypical samples and emphasizes that the direction of logits serves as a more reliable indicator of model confidence. In the blue dashed box of Fig. 1, we provide an example that illustrates the impact of TAL on an atypical sample. The logit vector could be changed from $\boldsymbol{f}_2$ to $\boldsymbol{f}_1$, indicating that the scores obtained with $\hat{\boldsymbol{f}}$ for both "horse"

and "human" become nearly equal. This alignment better aligns with human perception, highlighting the effectiveness of TAL in improving model confidence estimation.

The proposed TAL approach is model-agnostic, making it easily applicable to models with various architectures, such as CNN [12] and ViT [6]. Experimental results on benchmark datasets, including CIFAR10, CIFAR100, and ImageNet, demonstrate the superiority of TAL over existing failure detection methods. Specifically, on CIFAR100, our method achieves a significant improvement of more than 5% in terms of the Area Under the Risk-Coverage Curve (AURC) compared to the state-of-the-art method [49].

In summary, the main contributions of this paper are as follows:

- We propose a new insight that the overconfidence might stem from the presence of atypical samples, whose labels fail to accurately describe the images. This forces the models to conform to these imperfect labels during training, resulting in unreliable confidence scores.

- In order to mitigate the issue of overfitting on atypical samples, we introduce the Typicalness-Aware Learning (TAL), which enables the identification and separate optimization of typical and atypical samples, thereby alleviating the problem of overconfidence.

- Extensive experiments demonstrate the effectiveness and robustness of TAL. Besides, TAL has no structural constraints to the target model and is complementary to other existing failure detection methods.

## 2 Background and Preliminary

Prior to introducing our method, we present the background of Failure Detection (FD). Additionally, we highlight the distinctions between failure detection and two closely related concepts: Confidence Calibration (CC) and Out-of-Distribution detection (OoD-D).

**Failure Detection.** Failure detection (FD) [18, 26, 49, 50] aims to differentiate between correct and incorrect predictions by utilizing the ranking of their confidence levels. In particular, a confidence-rate function $\kappa(\cdot)$ is employed to assess the confidence level of each prediction. High-confidence predictions are accepted, while low-confidence predictions are rejected. By using a predetermined threshold $\delta \in \mathbb{R}^+$, users can make informed decisions based on the following function $g$:

$$g(\boldsymbol{x}) = \begin{cases} \text{accept} & \text{if } \kappa(\boldsymbol{x}) \geq \delta, \\ \text{reject} & \text{otherwise.} \end{cases} \tag{2}$$

where $\kappa(\cdot)$ denotes a confidence-rate function, such as the maximum softmax probability.

**Failure Detection vs. Confidence Calibration.** Confidence calibration (CC) [33, 23, 27, 45] primarily emphasizes the alignment of predicted probabilities with the actual likelihood of correctness, rather than explicitly detecting failed predictions as in FD. The goal of CC is to ensure that the predictive confidence is indicative of the true probability of correctness:

$$P\left(\hat{y} = y \mid \hat{p} = p^*\right) = p^*, \forall p^* \in [0, 1]. \tag{3}$$

This implies that when a model predicts a set of inputs $x$ to belong to class $y$ with a probability $p^*$, we would expect approximately $p^*$ of those inputs to truly belong to class $y$.

However, as observed by [49], models calibrated with CC algorithms do not perform well in FD. Traditional metrics used to evaluate CC, such as the Expected Calibration Error (ECE [28]), do not accurately reflect performance in FD scenarios. Instead, alternative metrics like the Area Under the Risk-Coverage Curve (AURC [7]) and the Area Under the Receiver Operating Characteristic Curve (AUROC [2]) are recommended for assessing FD performance.

**Failure Detection vs. Out-of-Distribution Detection.** While both Out-of-Distribution Detection (OoD-D) and failure detection tasks aim to enhance confidence reliability, they have distinct objectives, as depicted in Fig. 2. OoD-D focuses on rejecting predictions of semantic shift while accepting in-distribution predictions. However, it does not explicitly address the rejection of cases affected by covariate shifts. Additionally, through empirical observations in Sec. 4, we find that OoD-D methods are not well-suited for the Failure Detection task.

**Traditional Failure Detection (Old FD) vs. New Failure Detection (New FD).** Traditional failure detection methods [49, 50, 26] primarily focus on assessing the accuracy of predictions for in-distribution data. They also evaluate the discriminative performance of distinguishing correct and

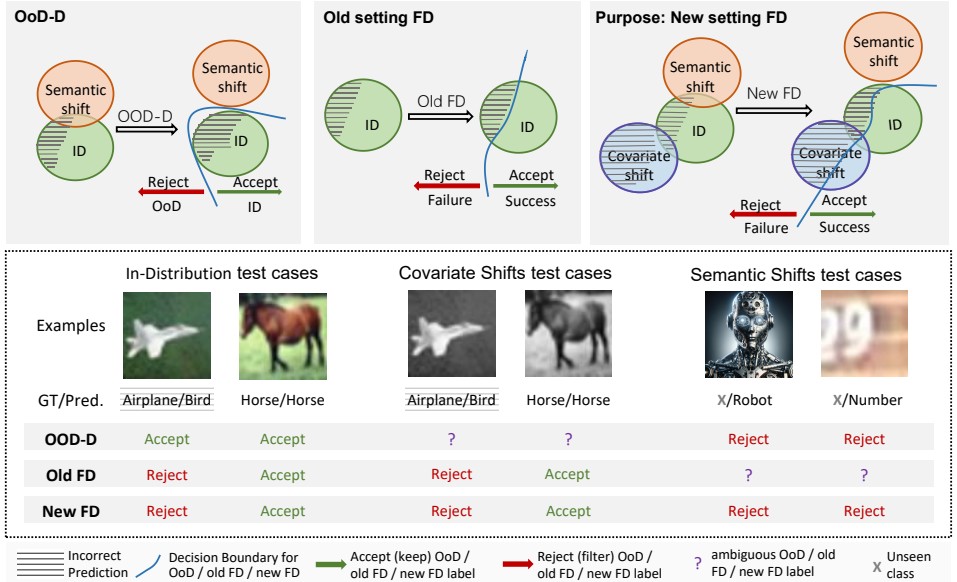

Figure 2: The differences between closely related tasks. The blue curve represents the decision boundary, and the shaded area in the figure indicates incorrect predictions. (a) illustrates the objective of OoD-D tasks to reject predictions with semantic shifts and accept in-distribution predictions, without concern for predictions with covariate shifts. (b) shows the old setting of FD tasks, accepting correct in-distribution predictions and rejecting incorrect out-of-distribution predictions. (c) displays the new setting of FD tasks, accepting correct in-distribution predictions and correct predictions with covariate shifts, while rejecting incorrect in-distribution predictions, incorrect predictions with covariate shifts, and predictions with semantic shifts. (d) illustrates examples of OoD-D, Old FD, and New FD tasks. A classifier trained on CIFAR10 [21] is evaluated on 6 images under a whole range of relevant distribution shifts: For instance, the 3rd and the 4th images in grayscale depict an airplane and a horse which encounter covariate shifts from that in the original CIFAR10. The 5th and the 6th images depict samples belonging to unseen categories with semantic shifts.

incorrect predictions for covariate shift data based on confidence scores. While these approaches address certain aspects of distribution shifts, they overlook the semantic shifts.

To address the limitations of Out-of-Distribution Detection (OoD-D) and traditional failure detection (Old FD) methods, [18] proposes a new setting called New FD. The objective of New FD is to accept correct predictions for both in-distribution and covariate shift samples, while rejecting incorrect predictions for all possible failures, including in-distribution, covariate shift, and semantic shift samples. Compared to OoD-D and Old FD, it enables more effective decision-making in real world.

## 3   Method

In this section, we present our proposed strategy called Typicalness-Aware Learning (TAL), as shown in Fig. 3. First, in Sec. 3.1, we address the shortcomings of existing training objectives and identify overfitting of atypical samples as a potential cause of overconfidence. Next, in Sec. 3.2, we outline the methodology used to calculate the "typicalness" of samples, enabling selective optimization and mitigating the negative impact of atypical samples. Finally, in Sec. 3.3, we introduce the TAL strategy, which incorporates the computed typicalness values for individual samples, resulting in improved performance for the failure detection task.

### 3.1   Revisit the Cross-entropy Loss

In Eq. (1), the optimization of the cross-entropy loss involves either increasing the magnitude of the logits or aligning them better with the labels. However, LogitNorm [40] researchers have observed that as the training progresses and the model becomes more accurate in classifying samples, it tends to generate significantly larger logit magnitudes, leading to overconfidence. To address this issue,

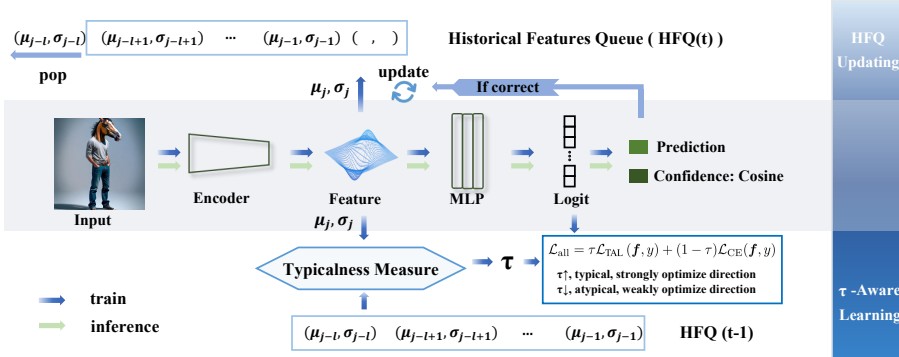

Figure 3: The framework of TAL. During training, statistical information (mean $\mu_j$ and variance $\sigma_j$) of features from correct predictions updates the Historical Features Queue (HFQ) at time-step t. The typicalness measure $\tau$ is calculated by comparing these statistics between the current batch and the HFQ. This $\tau$ influences the overall loss calculation, guiding the model to differentiate between atypical and typical samples. In the inference phase, TAL operates similarly to a model trained with conventional cross-entropy. Confidence is derived from the cosine similarity of the predicted logit direction, emphasizing our approach of using direction as a more reliable confidence metric. The framework distinguishes between typical (high $\tau$) and atypical (low $\tau$) samples, influencing the optimization process accordingly.

LogitNorm introduces a constant magnitude T in Eq. (4) to mitigate the problem:

$$\mathcal{L}_{\text{logit\_norm}}\left(\boldsymbol{f}, y\right) = -\log \frac{e^{\hat{\boldsymbol{f}}_y * \text{T}}}{\sum_{i=1}^{k} e^{\hat{\boldsymbol{f}}_i * \text{T}}}. \tag{4}$$

By keeping the magnitude constant in Eq. (4), the model places greater emphasis on producing features that align more closely with the target label in terms of direction, in order to minimize the training loss. However, as shown in Fig. 1, the presence of atypical samples with ambiguous content and labels may still cause overconfidence. Hence, it is imperative to devise a method that allows for the differentiation and separate optimization of typical and atypical samples.

Drawing inspiration from the cognitive process of human decision-making, it is reasonable to distinguish between typical and atypical samples by leveraging the knowledge acquired during the training phase. This approach allows us to effectively mitigate the negative effects of atypical samples, thereby preventing the occurrence of erroneous overconfidence.

## 3.2 Distinguish Typical and Atypical Samples

To differentiate typical samples from atypical ones, we introduce a method for evaluating typicalness and implement typicalness-aware learning (TAL). This approach entails calculating the mean and variance of the feature representations. Specifically, we calculate the mean and variance of each sample's feature channels based on insights from CORES [31]. The insight stems from the observation that in-distribution samples show larger magnitudes (mean) and variations (variance) in convolutional responses across channels compared to OoD samples, which are a type of atypical sample. The mean response of OOD samples is smaller than correct ID samples, as shown in Fig. 5(a). These statistical characteristics are subsequently compared to a set of historical data, representing typical samples, stored in a structured queue known as the "Historical Feature Queue" (HFQ). By comparing the statistical features of the input samples to those in the HFQ, we can quantify their typicalness.

**Initialize and update HFQ.** We commence the process by initializing the HFQ, denoted as $Q$, with a predetermined size equivalent to the number of samples in the training dataset. This structured queue is responsible for retaining the mean and variance of feature representations for typical samples identified throughout the training phase.

To establish the initial state of the queue, we do not adopt the model trained from scratch. Instead, we employ a model that has been trained for a few epochs, corresponding to a small portion ($\lambda$) of the total training epochs. This approach ensures the quality of the queue during the early stages of training. In this study, we set $\lambda = 0.05$, which corresponds to 5% of the total training duration.

During this initialization phase, for each batch of data, we calculate the mean ($\mu$) and variance ($\sigma^2$) of the feature vectors for each correctly predicted sample. Each sample in the queue stores its statistical features, denoted as follows, given a prediction $\hat{y}$ and the ground truth label $y$:

$$Q = \{(\mu_i, \sigma_i^2) \mid \hat{y}_i = y\} \tag{5}$$

where $\mu_i$ and $\sigma_i^2$ represent the mean and variance of the feature vectors of the $i$-th sample, respectively.

Once initialized, the statistics (mean and variance) of accurately predicted samples in each batch are directly added to the queue, as shown in Fig. 3. The queue has a fixed length of 20,000, and the ablation study is provided in Sec 4.3. The queue is updated using a First-In-First-Out (FIFO) approach, guaranteeing that it preserves a representative assortment of typical samples observed throughout the training process, while also adapting to the evolving data distributions. We empirically find this simple strategy works well in our experiments.

**Typicalness assessment.** To evaluate the typicalness $\tau$ of a new sample, we first calculate the mean ($\mu_{new}$) and variance ($\sigma_{new}^2$) of its features $\boldsymbol{f}$. Subsequently, we compute the $L2$ distance $d$ between the feature distribution of the new sample, represented by $\mu_{new}$ and $\sigma_{new}^2$, and the distributions of the features stored in the HFQ, denoted as $(\mu_j, \sigma_j^2) \in Q$. Finally, we normalize the resulting distance using min-max normalization to obtain the typicalness $\tau$.

$$d = \min_{(\mu_j, \sigma_j^2) \in Q} W((\mu_{new}, \sigma_{new}^2), (\mu_j, \sigma_j^2)), \tag{6}$$

$$\tau = 1 - \frac{d - d_{min}}{d_{max} - d_{min}}. \tag{7}$$

Where $d_{min}$ and $d_{max}$ represent the minimum and maximum distances of samples in the batch. Eq. (7) normalizes the value of $\tau$ within the range of $[0, 1]$, then $\tau$ can serve as a indicator of sample typicalness. A high $\tau$ value suggests that the sample is highly typical compared to the historical data. Conversely, a low $\tau$ value indicates an atypical or anomalous sample.

### 3.3 Typicalness-Aware Learning

Sec. 3.1 highlights the potential negative impact of atypical samples on the training process. Building upon the insights provided in Sec. 3.2, we now introduce Typicalness-Aware Learning (TAL) in this section. TAL leverages the typicalness $\tau$ to distinguish between typical and atypical samples during the optimization process. This approach aims to mitigate the issue of overconfidence that arises from the presence of atypical samples.

**The training objective of TAL.** The training objective of TAL is defined by incorporating an additional loss term $\mathcal{L}_{\text{TAL}}$. This is achieved by modifying the LogitNorm equation, denoted as Eq. (4), to Eq. (8) where the samples $\boldsymbol{x}$ are assigned with dynamic magnitudes $T(\tau)$ based on typicalness $\tau$.

$$\mathcal{L}_{\text{TAL}}(\boldsymbol{f}, y) = -\log \frac{e^{\hat{\boldsymbol{f}}_{\boldsymbol{y}} * T(\tau)}}{\sum_{i=1}^{k} e^{\hat{\boldsymbol{f}}_i * T(\tau)}}. \tag{8}$$

**Dynamic magnitude $T(\tau)$.** Given the upper bound $T_{\max}$ and the lower bound $T_{\min}$, the dynamic magnitude $T(\tau)$ can be obtained via:

$$T(\tau) = T_{\min} + (1 - \tau) \times (T_{\max} - T_{\min}), \tag{9}$$

where we empirically set $T_{\max}$ and $T_{\min}$ to 10 and 100, and they perform well on different benchmarks. The ablation study on different values is shown in Sec. 4.3.

Specifically, in Eq. (9), a *smaller* magnitude $T(\tau)$ will be assigned to *typical* samples with large $\tau$, and a *larger* magnitude $T(\tau)$ will be assigned to *less typical* samples with smaller $\tau$, enabling different treatments for typical/atypical samples. In this manner, for atypical samples, a higher value of $T(\tau)$ reduces the influence that pulls them towards the label direction. This helps prevent their logit directions from being excessively optimized.

In other words, the inverse proportionality between $T(\tau)$ and $\tau$ encourages the model to yield directions of $\boldsymbol{f}$ that are well-aligned with the labels for the typical samples with large $\tau$ by assigning a small magnitude $T(\tau)$. Conversely, for atypical samples with small $\tau$, the directions are not required

to be as precise as the typical ones as the current $T(\tau)$ is large, to mitigate the adverse impacts brought by the ambiguous label. To this end, *the direction $\hat{\boldsymbol{f}}$ can serve as a more reliable indicator of the model confidence*.

**The overall optimization.** Fig. 3 illustrates the TAL framework, showing both training and inference processes. During the training process, we utilize both the proposed TAL loss $\mathcal{L}_{\text{TAL}}$ and cross-entropy loss $\mathcal{L}_{\text{CE}}$ as it exhibits stronger feature extraction capabilities than LogitNorm [40]. The overall loss is:

$$\mathcal{L}_{\text{all}} = \tau \mathcal{L}_{\text{TAL}}\left(\boldsymbol{f}, y\right) + (1 - \tau)\mathcal{L}_{\text{CE}}(\boldsymbol{f}, y) \tag{10}$$

The TAL loss, denoted as $\mathcal{L}_{\text{TAL}}$, is utilized to optimize the directions of reliable typical samples with large typicalness $\tau$, as well as potentially some atypical samples with small $\tau$.

The CE loss (not only relying on the CE loss, as it is regulated by $\tau$) for atypical samples enables the optimization of both direction and magnitude. This may help reduce the adverse effects of atypical samples on the direction, enhancing the reliability of direction as a confidence indicator. This ensures that the optimization force on the logit directions of atypical samples is weaker compared to that of typical samples. The inference process does not involve the calculation of typicalness, and the only difference from the normal inference process is that our method uses Cosine as the confidence score.

To summarize, our proposed approach, as indicated in Eq. (10), enables models to selectively and adaptively optimize typical and atypical samples according to their typicalness values. This strategy enhances the reliability of feature directions as indicators of model confidence, ultimately improving the performance on failure detection task.

# 4 Experiments

To evaluate the effectiveness of the proposed Typicalness-Aware Learning (TAL) strategy, we conduct extensive experiments on various datasets, network architectures, and failure detection (FD) settings. More details such as the training configuration can be found in Appendix A.

**Datasets and models.** We first evaluate on the small-scale CIFAR-100 [21] dataset with SVHN [11] as its out-of-distribution (OOD) test set. To demonstrate scalability, we further conduct experiments on large-scale ImageNet [5] using ResNet-50, with Textures [3] and WILDS [20] serving as OOD data. For CIFAR-100, we verify TAL's effectiveness across diverse architectures including ResNet [14], WRNet [46], DenseNet [16], and the transformer-based DeiT-Small [39]. Detailed experimental results are provided in Appendix C.

**Three settings.** We evaluate TAL under three different settings: Old FD setting, OOD detection setting, and New FD setting (detailed in Section 2). While Old FD distinguishes between correct and incorrect in-distribution predictions, and OOD detection identifies out-of-distribution samples, our New FD setting aims to separate correctly predicted in-distribution samples from both mis-classified and out-of-distribution samples. We maintain a 1:1 ratio between in-distribution and out-of-distribution samples in testing, and also report results for Old FD and OOD detection settings for completeness.

**Baselines.** We compare our proposed TAL method against classical Maximum Softmax Probability (MSP), MaxLogit[15], Cosine[48], Energy [24], Entropy [34], Mahalanobis [4], Gradnorm [17], SIRC [41] and recent LogitNorm [40], OpenMix [50] and (FMFP) [49]. It is worth noting that FMFP focuses on improving accuracy for failure detection.

**Evaluation metrics.** To comprehensively assess the performance of TAL in failure detection, we adopt three widely recognized evaluation metrics [18, 49, 9], including Area Under the Risk-Coverage Curve (AURC), Area Under the Receiver Operating Characteristic Curve (AUROC), False Positive Rate at 95% True Positive Rate (FPR95).

## 4.1 Comparisions with the State-of-the-art on CIFAR

**Evaluation with CNN-based architectures.** As shown in Tab. 1, our TAL strategy outperforms existing methods in New FD settings. Here are the key observations: 1) OoD methods like Energy and LogitNorm do not achieve satisfactory performance in the FD task. Please refer to Appendix B for explanation. 2) TAL consistently surpasses the baseline MSP and MaxLogit across various network architectures by a large margin. 3) In terms of comparison with FMFP, the prior SOTA

method in the field of FD, our analysis reveals that TAL is complementary to the FMFP method and exhibits superior performance when combined with it. 4) Regarding Openmix, the metrics presented demonstrate that it falls short when compared to TAL.

Table 1: Evaluation results of the proposed TAL on CIFAR100.

| Architecture | Method | Old setting FD | | | OOD Detection | | | New setting FD | | | ID-ACC |
|---|---|---|---|---|---|---|---|---|---|---|---|
| | | AURC↓ | FPR95↓ | AUROC↑ | AURC↓ | FPR95↓ | AUROC↑ | AURC↓ | FPR95↓ | AUROC↑ | |
| | | | | | CIFAR100 vs. SVHN | | | | | | |
| ResNet110 [14] | MSP[15] | 99.83 | 67.49 | 84.07 | 293.44 | 83.41 | 74.55 | 376.42 | 66.92 | 84.00 | 72.01 |
| | Cosine [48] | 96.53 | 65.15 | 84.42 | 271.13 | 78.30 | 79.31 | 361.87 | 56.23 | 86.93 | 72.01 |
| | Energy [24] | 135.85 | 74.66 | 77.20 | 275.39 | 83.18 | 77.78 | 387.44 | 66.96 | 83.21 | 72.01 |
| | MaxLogit[15] | 133.19 | 72.33 | 77.96 | 275.85 | 82.53 | 77.73 | 385.81 | 65.08 | 83.56 | 72.01 |
| | Entropy [34] | 100.05 | 66.28 | 84.12 | 287.62 | 81.20 | 75.93 | 373.49 | 61.33 | 84.73 | 72.01 |
| | Mahalanobis [4] | 114.21 | 73.48 | 80.41 | 263.49 | 72.70 | 80.55 | 368.55 | 58.74 | 85.74 | 72.01 |
| | Gradnorm [17] | 369.86 | 98.82 | 35.30 | 490.21 | 98.17 | 49.26 | 679.48 | 98.69 | 42.76 | 72.01 |
| | SIRC [41] | 100.56 | 66.37 | 84.01 | 287.93 | 81.03 | 75.90 | 374.12 | 61.29 | 84.65 | 72.01 |
| | LogitNorm [40] | 125.59 | 72.87 | 79.71 | 235.50 | 73.23 | 83.35 | 356.88 | 55.80 | 87.80 | 70.34 |
| | OpenMix [50] | 85.66 | 63.82 | 85.25 | 342.16 | 87.03 | 69.27 | 406.80 | 70.37 | 80.25 | 73.68 |
| | TAL | 90.60 | 64.84 | 85.36 | 259.64 | 76.37 | 80.28 | 347.72 | 54.39 | 87.89 | 72.45 |
| | FMFP [49] | 69.83 | 62.17 | 87.15 | 284.13 | 81.77 | 74.98 | 345.37 | 62.99 | 84.86 | 75.18 |
| | TAL w/ FMFP | 73.16 | 64.82 | 85.51 | 245.62 | 78.61 | 81.59 | 320.73 | 55.22 | 88.48 | 75.59 |
| WRNet [46] | MSP[15] | 57.60 | 60.13 | 87.47 | 264.67 | 80.04 | 79.41 | 321.39 | 57.26 | 87.23 | 78.55 |
| | Cosine [48] | 56.68 | 58.65 | 87.56 | 279.53 | 81.60 | 77.91 | 334.04 | 57.96 | 86.08 | 78.55 |
| | Energy [24] | 66.33 | 65.64 | 84.73 | 257.78 | 78.93 | 80.83 | 323.27 | 58.70 | 87.07 | 78.55 |
| | MaxLogit[15] | 64.85 | 62.70 | 85.36 | 258.57 | 79.35 | 80.67 | 322.42 | 56.89 | 87.29 | 78.55 |
| | Entropy [34] | 59.31 | 62.78 | 86.80 | 260.92 | 79.50 | 80.26 | 320.66 | 56.92 | 87.40 | 78.55 |
| | Mahalanobis [4] | 76.43 | 72.99 | 81.44 | 231.16 | 61.11 | 85.74 | 312.53 | 50.49 | 88.73 | 78.55 |
| | Gradnorm [17] | 134.85 | 76.20 | 68.51 | 305.28 | 75.01 | 75.57 | 405.60 | 65.57 | 77.46 | 78.55 |
| | SIRC [41] | 59.57 | 63.38 | 86.70 | 259.93 | 77.99 | 80.50 | 320.35 | 56.93 | 87.45 | 78.55 |
| | LogitNorm [40] | 78.86 | 65.56 | 82.68 | 204.45 | 58.98 | 89.10 | 294.87 | 41.88 | 92.03 | 77.15 |
| | OpenMix [50] | 50.05 | 56.36 | 88.73 | 251.49 | 74.33 | 81.16 | 304.35 | 51.75 | 88.59 | 79.52 |
| | TAL | 55.41 | 58.43 | 88.41 | 245.36 | 79.48 | 81.29 | 303.18 | 54.62 | 89.22 | 78.14 |
| | FMFP [49] | 41.60 | 56.65 | 89.50 | 245.10 | 75.71 | 81.42 | 290.23 | 55.63 | 88.89 | 81.07 |
| | TAL w/ FMFP | 44.21 | 58.77 | 88.40 | 224.56 | 71.32 | 85.31 | 278.28 | 47.14 | 91.05 | 80.98 |
| DenseNet [16] | MSP[15] | 79.06 | 63.62 | 85.75 | 257.69 | 77.86 | 79.77 | 333.86 | 57.58 | 87.64 | 74.62 |
| | Cosine [48] | 82.36 | 63.65 | 84.72 | 250.14 | 77.01 | 81.48 | 332.18 | 53.82 | 88.28 | 74.62 |
| | Energy [24] | 108.80 | 72.79 | 78.73 | 240.11 | 75.65 | 82.81 | 341.87 | 58.95 | 86.94 | 74.62 |
| | MaxLogit[15] | 105.98 | 70.20 | 79.60 | 240.06 | 75.65 | 82.86 | 339.68 | 56.69 | 87.39 | 74.62 |
| | Entropy [34] | 79.94 | 65.41 | 85.46 | 250.68 | 75.54 | 81.34 | 330.88 | 53.97 | 88.30 | 74.62 |
| | Mahalanobis [4] | 159.95 | 95.30 | 64.54 | 208.53 | 54.83 | 89.23 | 358.90 | 63.29 | 84.93 | 74.62 |
| | Gradnorm [17] | 249.85 | 94.73 | 49.88 | 310.37 | 80.45 | 73.68 | 483.52 | 83.21 | 68.79 | 74.62 |
| | SIRC [41] | 80.21 | 65.41 | 85.37 | 249.73 | 74.48 | 81.58 | 330.63 | 53.32 | 88.36 | 74.62 |
| | LogitNorm [40] | 109.80 | 73.28 | 78.90 | 216.63 | 60.21 | 87.27 | 330.07 | 48.43 | 89.50 | 73.80 |
| | OpenMix [50] | 64.11 | 59.52 | 87.29 | 267.67 | 77.90 | 78.27 | 328.27 | 58.83 | 86.79 | 77.03 |
| | TAL | 73.28 | 60.90 | 86.51 | 250.78 | 75.46 | 81.35 | 325.70 | 52.05 | 88.73 | 75.14 |
| | FMFP [49] | 56.67 | 62.45 | 87.72 | 258.04 | 77.11 | 79.31 | 314.57 | 59.81 | 87.37 | 77.96 |
| | TAL w/ FMFP | 60.52 | 63.65 | 86.57 | 224.62 | 63.43 | 86.01 | 296.75 | 45.10 | 90.93 | 77.85 |

## 4.2 Comparisions with the Baseline on ImageNet.

Table 2: Evaluation results of the proposed TAL on ImageNet.

| Architecture | Method | Old setting FD | | | OOD Detection | | | New setting FD | | | ID-ACC |
|---|---|---|---|---|---|---|---|---|---|---|---|
| | | AURC↓ | FPR95↓ | AUROC↑ | AURC↓ | FPR95↓ | AUROC↑ | AURC↓ | FPR95↓ | AUROC↑ | |
| | | | | | Imagenet vs. Textures | | | | | | |
| ResNet50 [13] | MSP [15] | 72.73 | 63.95 | 86.18 | 301.27 | 46.01 | 87.21 | 351.26 | 49.64 | 86.99 | 76.13 |
| | Cosine [48] | 102.98 | 69.93 | 79.49 | 298.35 | 50.64 | 87.54 | 359.74 | 54.43 | 86.17 | 76.13 |
| | Energy [24] | 118.66 | 76.33 | 75.81 | 279.16 | 35.64 | 90.47 | 351.93 | 43.69 | 87.74 | 76.13 |
| | MaxLogit [15] | 113.35 | 72.11 | 77.29 | 278.52 | 34.1 | 90.57 | 349.3 | 41.59 | 88.1 | 76.13 |
| | Entropy [34] | 74.61 | 67.07 | 85.48 | 292.54 | 38.3 | 88.92 | 344.73 | 43.95 | 88.27 | 76.13 |
| | Mahalanobis [4] | 208.22 | 96.19 | 54.23 | 288.17 | 57.61 | 86.51 | 397.18 | 65.34 | 80.22 | 76.13 |
| | Residual [41] | 238.18 | 97.01 | 49.0 | 316.1 | 57.77 | 83.89 | 431.12 | 65.55 | 77.12 | 76.13 |
| | Gradnorm [17] | 206.99 | 89.66 | 57.88 | 272.83 | 30.21 | 91.55 | 385.97 | 42.45 | 84.89 | 76.13 |
| | SIRC [41] | 72.91 | 63.67 | 86.11 | 295.13 | 38.88 | 88.53 | 346.42 | 43.82 | 88.03 | 76.13 |
| | TAL | 64.66 | 64.93 | 87.11 | 290.5 | 47.66 | 87.51 | 338.45 | 50.11 | 88.29 | 76.43 |
| | TAL+SIRC | 64.55 | 63.66 | 87.15 | 288.23 | 46.91 | 87.88 | 336.56 | 49.68 | 88.35 | 76.43 |
| | | | | | Imagenet vs. WILDS | | | | | | |
| ResNet50 [13] | MSP [15] | 72.73 | 63.95 | 86.18 | 272.93 | 59.27 | 87.72 | 326.85 | 60.19 | 87.42 | 76.13 |
| | Cosine [48] | 102.98 | 69.93 | 79.49 | 255.91 | 68.67 | 89.85 | 326.97 | 68.92 | 87.81 | 76.13 |
| | Energy [24] | 118.66 | 76.33 | 75.81 | 235.88 | 37.67 | 94.22 | 318.89 | 45.27 | 90.60 | 76.13 |
| | MaxLogit [15] | 113.35 | 72.11 | 77.29 | 237.23 | 38.93 | 93.97 | 317.46 | 45.46 | 90.69 | 76.13 |
| | Entropy [34] | 74.61 | 67.07 | 85.48 | 259.01 | 51.20 | 90.44 | 316.26 | 54.32 | 89.47 | 76.13 |
| | Mahalanobis [4] | 208.22 | 96.19 | 54.23 | 264.11 | 77.17 | 88.20 | 382.46 | 80.91 | 81.51 | 76.13 |
| | Residual [41] | 238.18 | 97.01 | 49.00 | 282.46 | 81.30 | 85.09 | 409.89 | 84.39 | 77.99 | 76.13 |
| | Gradnorm [17] | 206.99 | 89.66 | 57.88 | 237.31 | 25.37 | 94.84 | 363.03 | 38.02 | 87.56 | 76.13 |
| | SIRC [41] | 72.91 | 63.67 | 86.11 | 267.33 | 52.17 | 89.03 | 322.41 | 54.43 | 88.46 | 76.13 |
| | TAL (ours) | 64.66 | 64.93 | 87.11 | 232.11 | 40.97 | 94.28 | 288.67 | 45.55 | 92.91 | 76.43 |

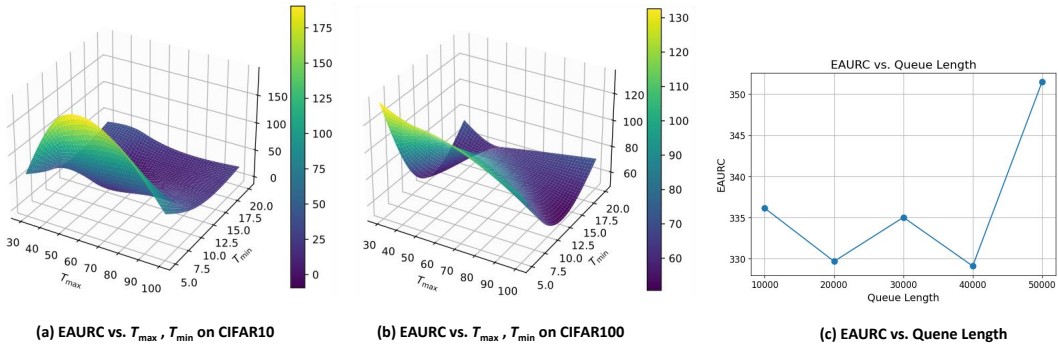

Figure 4: (a) and (b) is the ablation study of $T_{\min}$, $T_{\max}$. And (c) is the ablation study on the length of the Historical Feature Queue.

To showcase the scalability of our approach, we present the results on ImageNet in Table 2. It is obvious that our TAL strategy consistently enhances the failure detection performance of the baseline method, significantly improving the reliability of confidence. Notably, TAL reduces the AURC by 3.7 and 11.6 points, indicating a better overall performance in distinguishing between correct and incorrect predictions. It is worth noting that TAL achieves these impressive improvements while maintaining a comparable accuracy to the MSP baseline.

Additionally, we present the Risk-Coverage (RC) curves (Fig. 5(c)) for both old and new FD task settings on ImageNet. The comparison between TAL and baseline RC curves demonstrates the effectiveness of our method. Fig. 5(d) further visualizes typical and atypical data examples. For the fish category in ImageNet, typical data includes common fish images, while atypical data comprises both rare fish images from ImageNet and out-of-distribution samples.

## 4.3 Ablation Study

**The ablation study of key components.** We conduct experimental ablations on the components of our TAL loss with CIFAR100. The results are summarized in Table 3. With the dynamic magnitude $T(\tau)$ strategy, we achieve substantial enhancements in Failure Detection performance, which manifests the effectiveness of integrating typicalness-aware strategies into training approaches.

Table 3: Ablation of the key components. **Best** are bolded and second best are underlined. AURC and EAURC are multiplied by $10^3$, the remaining metrics are percentages except ACC. "Fixed T" means the dynamic magnitude $T(\tau)$ in TAL is not adopted.

| Method | Setting | AURC ↓ | EAURC ↓ | AUROC ↑ | FPR95 ↓ | TNR95 ↑ | AUPR-Success ↑ | AUPR-Error ↑ | ACC ↑ |
|---|---|---|---|---|---|---|---|---|---|
| Fixed T | Old FD | 108.46 | 58.81 | 83.87 | 62.76 | 37.24 | 92.23 | 68.35 | 0.70 |
| | New FD | 355.62 | **69.67** | 88.74 | 53.45 | 46.52 | **83.78** | 92.51 | 0.70 |
| Fixed T + Cross entropy | Old FD | 99.60 | 55.63 | 83.57 | 65.65 | 34.35 | 92.81 | 66.00 | 0.72 |
| | New FD | 362.77 | 85.41 | 86.66 | 57.00 | 43.00 | 80.57 | 91.10 | 0.72 |
| TAL loss (Dynamic T) + Cross entropy | Old FD | **94.33** | **49.43** | **85.58** | **61.24** | **38.69** | **93.56** | **68.70** | **0.72** |
| | New FD | **351.49** | 72.69 | **88.92** | **47.44** | **52.46** | 83.03 | **92.86** | **0.72** |

**The impacts of $T_{\min}$ and $T_{\max}$.** We perform an ablation study using ResNet110 on the CIFAR10 and CIFAR100 datasets to examine the impact of $T_{\min}$ and $T_{\max}$ on failure detection performance. Fig. 4 (a) and Fig. 4 (b) present the experimental results of the failure detection metric EAURC for CIFAR10 and CIFAR100, respectively. Darker regions in the figures correspond to lower values of the metric, indicating superior failure detection performance. The findings suggest that while $T_{\min}$ should not be set too small, a moderate increase in $T_{\max}$ can enhance failure detection capabilities.

**The effects of the length of Historical Feature Queue.** We conduct an ablation study on the CIFAR100 dataset to examine the impact of queue length on failure detection performance. The original CIFAR100 dataset consists of 50,000 training images, with 5,000 images reserved for validation and the remaining 45,000 images used for training. The results, depicted in Figure 4 (c), demonstrate that queue lengths ranging from 10,000 to 50,000 yield similar failure detection performance. However, when the queue length exceeds 50,000, there is a noticeable decline in failure detection performance.

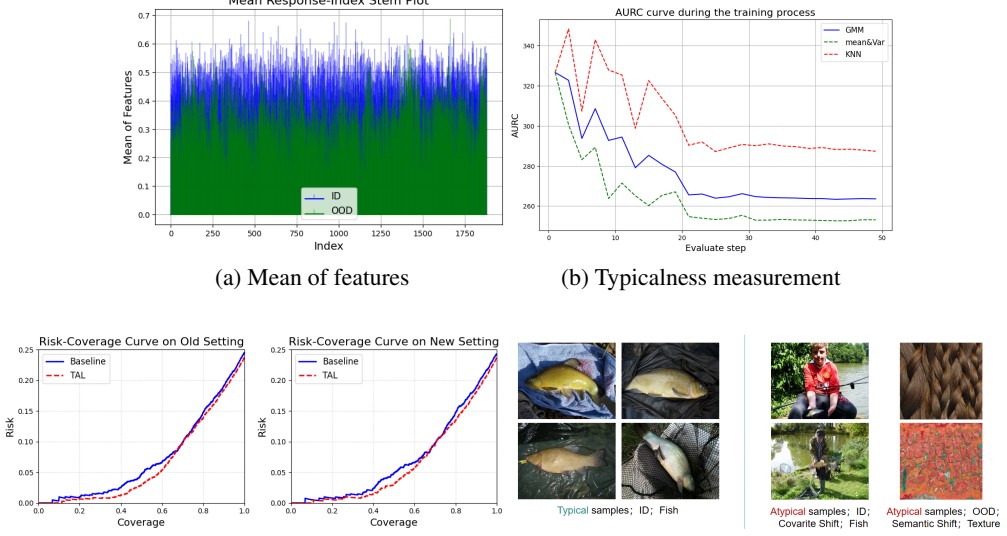

(a) Mean of features          (b) Typicalness measurement

(c) RC curves          (d) Typical and atypical examples

Figure 5: (a) Comparison of the Mean of Features between ID and OOD; (b) Comparison of different methods for measuring typicality; (c) The Risk-Coverage curves on old and new setting FD tasks; (d) Examples of typical and atypical examples.

**Ablation of Typicality Measures.** As depicted in Fig. 5 (b), we have conduced extra ablation experiments with K-nearest neighbor (KNN) distance and Gaussian Mixture Models (GMM) to assess typicality. These alternative measures did not enhance performance (lower AURC is preferable), thereby reinforcing the validity of our selection of mean/variance criteria.

## 5 Concluding Remarks

**Summary.** This paper introduces Typicalness-Aware Learning (TAL), a novel approach for mitigating overconfidence in DNNs and improving failure detection performance. The effectiveness of TAL can be attributed to a crucial insight: overconfidence in deep neural networks (DNNs) may arise when models are compelled to conform to labels that inadequately describe the image content of atypical samples. To address this issue, TAL leverages the concept of typicalness to differentiate the optimization of typical and atypical samples, thereby enhancing the reliability of confidence scores. Extensive experiments have been conducted to validate the effectiveness and robustness of TAL. We hope TAL can inspire new ideas for further enhancing the trustworthiness of deep learning models.

**Limitations.** The main contribution of TAL lies in recognizing the issue of overfitting atypical samples as a cause of overconfidence and proposing a comprehensive framework to tackle this problem. Given that the methods adopted in this work are simple yet effective, there is still potential for further improvement by incorporating more advanced designs, such as the methods for typicalness calculation and the dynamic magnitude generation. These are left as future work to be explored.

**Broader impacts.** As deep learning models become increasingly integrated into critical systems, from autonomous vehicles to medical diagnostics, the need for accurate and reliable confidence scores is paramount. TAL's ability to improve failure detection performance directly addresses this need, potentially leading to safer and more dependable AI systems.

## Acknowledgments

This work was supported by National Natural Science Foundation of China (grant No. 62376068, grant No. 62350710797), by Guangdong Basic and Applied Basic Research Foundation (grant No. 2023B1515120065), by Shenzhen Science and Technology Innovation Program (grant No. JCYJ20220818102414031).

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

# Appendix

## Overview

This is the supplementary material for our submission titled *Typicalness-Aware Learning for Failure Detection*. This material supplements the main paper with the following content:

- A. **More Experimental Details**
    - A.1. Baselines
    - A.2. Evaluation Metrics
    - A.3. Training Configuration
    - A.4. Test Configuration

- B. **The Reasons Why OoD method Performs Poor in FD**

- C. **Additional Results**

## A    More Experimental Details

### A.1    Baselines

We compare our proposed TAL method against eleven baseline approaches for failure detection. ① Maximum Softmax Probability (MSP), ② MaxLogit[15] and ③ Cosine[48] utilize the maximum softmax probability of $f$, the maximum logit ($f$) value and the cosine similarity between the $f$ and the corresponding label's one-hot vector as the confidence score, respectively. ④ Energy Score [24] uses the negative energy of the softmax output as the confidence score. ⑤ LogitNorm [40] normalizes the logits to a fixed magnitude to improve confidence score reliability. ⑥ Entropy [34] employs the entropy of the softmax distribution as an uncertainty measure. ⑦ Mahalanobis [4] computes confidence scores based on the Mahalanobis distance in the feature space. ⑧ Gradnorm [17] utilizes the gradient norm of the loss with respect to the model parameters as a measure of uncertainty. ⑨ OpenMix [50], a SOTA OOD detection method, leverages data augmentation techniques to enhance confidence score separation between in-distribution and out-of-distribution samples. ⑩ SIRC [41] augments softmax-based confidence scores with feature-agnostic information to better identify OOD samples while maintaining separation between correct and incorrect ID predictions. ⑪ Failure Misclassification Feature Propagation (FMFP) [49], a SOTA failure detection method, focuses on improving model accuracy and confidence score reliability through stochastic weight averaging (SWA) and sharpness-aware minimization (SAM). We also explore combining the proposed TAL with FMFP (⑫) to investigate their complementary effects.

### A.2    Evaluation Metrics

To comprehensively assess the performance of TAL in failure detection, we adopt nine widely recognized evaluation metrics [18, 49, 9], including Area Under the Risk-Coverage Curve (AURC), Excess Area Under the Risk-Coverage Curve (EAURC), Area Under the Receiver Operating Characteristic Curve (AUROC), False Positive Rate at 95% True Positive Rate (FPR95), True Negative Rate at 95% True Positive Rate (TNR95), Area Under the Precision-Recall curve of Suceess and Error (AUPR_Success and AUPR_Error). ① AURC [7]: Area Under the Risk-Coverage Curve, depicting the error rate as a function of confidence thresholds. ② EAURC [10]: Excess Area Under the Risk-Coverage Curve, evaluating the ranking ability of confidence scores. ③ AUROC [2]: Area Under the Receiver Operating Characteristic Curve, illustrating the trade-off between true positive rate (TPR) and false positive rate (FPR). ④: False Positive Rate at 95% True Positive Rate. ⑤: True Negative Rate at 95% True Positive Rate. ⑦ AUPR_Success[1] and ⑧ AUPR_Success represent two approximations for estimating the Area Under the Precision-Recall curve (AUPR), with AUPR_Success sampling thresholds from positive scores while AUPR_Error utilizes only scores of observed positive and negative instances as thresholds. ⑨: Test accuracy, providing a reference for overall model performance.

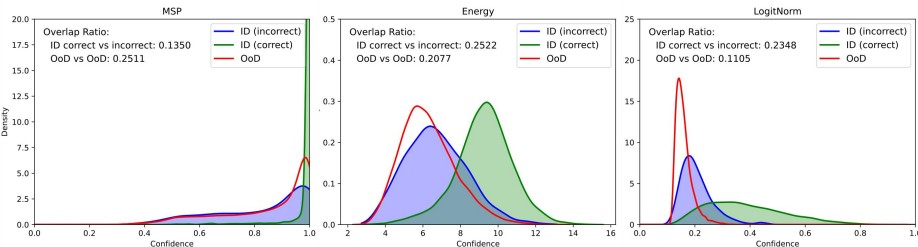

Figure 6: OOD-D methods lead to worse confidence separation between correct and wrong samples.

## A.3 Training Configuration

For experiments on the CIFAR [21], we employ an SGD optimizer with an initial learning rate of 0.1, a momentum of 0.9, and a weight decay of 0.0005. The models are trained for 200 epochs with a batch size of 256 on a single NVIDIA GeForce RTX 3090 GPU. Furthermore, we adopt a CosineAnnealingLR scheduler to adjust the learning rate during training. On ImageNet [5], we use the ResNet-50 architecture as our backbone. The models are trained for 90 epochs with an initial learning rate of 0.1 on a single NVIDIA A100. The learning rate is decayed by a factor of 0.1 every 30 epochs.

## A.4 Evaluation Configuration

To ensure a fair and robust evaluation, we conduct experiments with three independent training runs on CIFAR100, each using a different random seed. All baseline methods and our proposed TAL are evaluated using identical settings across these three runs. The final performance metrics reported in our results are averaged across these three sets of weights to account for training variance. For clarity of presentation, we multiply AURC values by $10^3$, while maintaining all other metrics in percentage form. This systematic evaluation approach allows us to make reliable comparisons between different methods while accounting for the inherent variability in deep learning model training.

For ImageNet experiments, we encountered several implementation challenges with some baseline methods. Specifically, FMFP failed to achieve competitive accuracy, while LogitNorm and OpenMix suffered from training instability and collapse. These issues might be attributed to the fact that these methods were not originally validated on ImageNet in their respective papers. While it might be possible to adapt these methods for ImageNet through extensive parameter tuning and modifications, such adaptations would require significant engineering effort and might deviate from the original methods. Therefore, we opted not to report results for these methods on ImageNet to maintain experimental integrity.

## B The Reasons Why OoD method Performs Poor in FD

It is interesting to note that in the Old Few-Shot Detection (FD) setting, most Out-of-Distribution (OoD) methods, such as Energy Score and LogitNorm, exhibit poor performance compared to baseline methods like Maximum Softmax Probability (MSP) and MaxLogit. This decline in performance can be attributed to the fact that while OoD methods effectively increase the confidence score gap between in-distribution and out-of-distribution samples, they inadvertently disrupt the natural confidence score hierarchy within the in-distribution samples. As a result, there is a greater overlap in the confidence distributions of correctly and incorrectly predicted in-distribution samples, as illustrated in Fig. 6. This observation highlights the importance of developing dedicated FD methods that can effectively distinguish between correct and incorrect predictions within the in-distribution data.

## C Additional Results

### Evaluation on Transformer-based architectures.

Considering the remarkable success of vision transformers as network architectures, it is crucial to incorporate a transformer-based network in our analysis and evaluate the effectiveness of our proposed

method. It is worth noting that the substantial disparities between ViT [6] and CNN architectures in terms of their design, feature representation, and learning mechanisms may pose challenges in directly applying methods that have proven effective on CNNs to ViT models.

**Implementation Details of ViT.** TAL can also be applied to Transformer architectures. However, it is important to note that certain CNN-based methods may not perform well on Transformer-based tasks. For example, the accuracy of OpenMix with default settings is significantly lower (approximately 0.20) compared to the baseline. This indicates that OpenMix, originally designed for CNNs, does not effectively detect failures when applied to ViT models without appropriate modifications and adjustments that consider the unique architectural characteristics of ViT.

Specifically, ViT learns the relationships between image patches through the Self-Attention mechanism, resulting in distinct feature representations and gradient flow patterns compared to CNNs. Moreover, ViT typically employs different normalization techniques, such as Layer Normalization, instead of the commonly used Batch Normalization in CNNs. These differences can impact the effectiveness of certain methods when applied to ViT. To successfully adapt these methods to ViT, appropriate modifications and adjustments may be necessary to account for the unique architectural characteristics of ViT.

To explore the performance of current popular failure detection methods on ViT models, we conducted experiments using the pre-trained DeiT-Small model ("deit_small_patch16_224") from the timm library. We employed the SGD optimizer with a base learning rate of 0.01, a weight decay of 0.0005, and a momentum of 0.9. The learning rate scheduler was set to CosineAnnealingLR, with the T_max parameter determined by total training epochs. The experiments were conducted on the CIFAR100 dataset, with a total of 25 training epochs and a batch size of 256.

**Experimental Results on ViT.** The experimental results shown in Tab. 4 and Tab. 5 validate the applicability of TAL to vision transformers. By prioritizing optimization with typical samples and concurrently relaxing the requirements for atypical samples, TAL effectively mitigates overconfidence and enhances failure detection performance.

Table 4: New FD Setting evaluation on CIFAR100 with ViT. Mean and standard deviations of Failure Detection performance on CIFAR benchmarks. The experimental results are reported over five epochs. **Best** are bolded and second best are underlined. AURC and EAURC are multiplied by $10^3$, the remaining metrics are percentages except ACC.

| Method | AURC ↓ | EAURC ↓ | AUROC ↑ | FPR95 ↓ | TNR95 ↑ | AUPR-Success ↑ | AUPR-Error ↑ | ACC ↑ |
|---|---|---|---|---|---|---|---|---|
| MSP [15] | 269.42±5.71 | 60.10±5.80 | 89.75±1.10 | 49.65±4.15 | 50.35±4.15 | 88.02±1.06 | 91.48±1.05 | 0.86±0.00 |
| LogitNorm [40] | 268.20±9.40 | 57.84±9.90 | 91.25±0.99 | 37.78±2.18 | 62.19±2.17 | 88.03±2.05 | 93.38±0.61 | 0.86±0.00 |
| **TAL** | **262.57±7.60** | **53.34±7.82** | **91.61±0.65** | **35.64±1.07** | **64.34±1.05** | **89.01±1.68** | **93.57±0.28** | **0.87±0.00** |
| FMFP [49] | 255.89±2.99 | 50.37±3.56 | 91.09±0.69 | 45.11±3.19 | 54.86±3.21 | 90.02±0.63 | 92.40±0.73 | 0.87±0.00 |
| **TAL w/ FMFP** | **246.07±2.21** | **39.96±2.89** | **93.17±0.76** | **31.84±2.53** | **68.15±2.46** | **91.84±0.60** | **94.41±0.68** | 0.87±0.00 |

Table 5: Old Setting evaluation on CIFAR100 with ViT. Mean and standard deviations of Failure Detection performance on CIFAR benchmarks. The experimental results are reported over five epochs. **Best** are bolded and second best are underlined. AURC and EAURC are multiplied by $10^3$, the remaining metrics are percentages except ACC.

| Method | Setting | AURC ↓ | EAURC ↓ | AUROC ↑ | FPR95 ↓ | TNR95 ↑ | AUPR-Success ↑ | AUPR-Error ↑ | ACC ↑ |
|---|---|---|---|---|---|---|---|---|---|
| MSP [15] | Old FD | 27.72±0.61 | 18.17±0.61 | 88.88±0.33 | 56.71±2.18 | 43.26±2.17 | 97.96±0.07 | 54.36±1.43 | 0.86±0.00 |
| LogitNorm [40] | Old FD | 27.46±0.32 | 17.54±0.19 | 89.16±0.26 | 56.99±1.32 | 42.98±1.35 | **98.03±0.02** | **54.92±0.93** | 0.86±0.00 |
| **TAL** | Old FD | 27.15±0.33 | 17.62±0.20 | **89.83±0.15** | **55.65±1.62** | **44.28±1.69** | **98.03±0.02** | 54.47±0.30 | **0.87±0.00** |
| FMFP [49] | Old FD | 22.58±0.31 | **14.30±0.48** | **90.11±0.41** | 54.35±3.27 | 45.62±3.27 | **98.41±0.05** | 54.67±2.80 | 0.87±0.00 |
| **TAL w/ FMFP** | Old FD | **21.02±0.32** | 14.55±0.21 | 88.79±0.54 | **53.00±2.43** | **46.99±2.35** | 98.15±0.07 | **54.88±1.90** | 0.87±0.00 |

