# OpenReview forum: "Typicalness-Aware Learning for Failure Detection"
_NeurIPS.cc/2024/Conference — NeurIPS 2024 poster_

### Official Review · Reviewer_yqWV · 2024-07-10

**Soundness:** 3
**Presentation:** 4
**Contribution:** 3
**Rating:** 7
**Confidence:** 4

**Summary:**

This paper identifies overfitting of atypical samples as a potential casue of overconfidence in DNN for Failure Detection.
The authors proposes a Typicalness-Aware Learning (TAL) approach that computes a typicalness score of each sample. TAL assigns dynamic logit magnitudes based on typicalness to allow flexible fitting of atypical samples while preserving reliable directions as confidence for atypical and typical samples.

**Strengths:**

1. This paper proposes a novel perspective - overconfidence may be caused by models being forced to conform to labels that fail to accurately describle the image content of atypical samples.
2. This paper introduces TAL which distinguishes between optimizing typical and atypical samples, thereby improving the reliability of confidence scores.
3. TAL is model-agnostic and can be readily applied to Transformers and CNNs, demontrating the strong adaptability of TAL.
4.  TAL is complementary to existing failure dectection methods and can be conbined for further performance gains.

**Weaknesses:**

1. The definition of atypical samples is vague, making it difficult to accurately determine the typicality of samples, which may affect the method's effectiveness. Please consider providing some mathematical definitions and visual examples.
2. There is room for improvement in the specific implementation, such as typically calculation and dynamic magnitude generation, by exploring more advanced designs.
3. The paper only compare failure detection task; the impact of TAL on classification accuracy is not analyzed ( accuracy is noly provided in the table but not discussed).
4. Minor error: "pred" and "GT" in Fig.2 should be swapped.

**Questions:**

1. In Section 3.2 "We empirically find this simple strately works well in our experiments." Please explain why the mean and variance can represent the features of a sample to measure typicality?
2. The confidence calibration methods aim to achieve more reliable confidence estimates, but fail in failure detection task. Could please explain that?
3. The OoD data and ID data are presented in equal proportions. How does the ration of them affect the results.

**Limitations:**

Discussed

---

> ### Author Rebuttal · Authors · 2024-08-07
>
> We thank the reviewer for the valuable comments and suggestions.
>
> **Q1**：The definition of atypical samples is vague, making it difficult to accurately determine the typicality of samples, which may affect the method's effectiveness. Please consider providing some mathematical definitions and visual examples.
>
> Thank you for your comments, and we have added the mathematical definition of typicality from the NeurIPS 2023 [ref1] in the revised version of our manuscript.
>
> Furthermore, we present the visual examples in Fig. 7 (b). These examples may aid in clarifying the distinction between typical and atypical samples.
>
> [ref1] Yuksekgonul, "Beyond Confidence: Reliable Models Should Also Consider Atypicality," NeurIPS 2023.
>
> **Q2**：There is room for improvement in the specific implementation, such as typically calculation and dynamic magnitude generation, by exploring more advanced designs.
>
> As depicted in Fig. 5 (b), we have conduced extra ablation experiments with K-nearest neighbor (KNN) distance and Gaussian Mixture Models (GMM) to assess typicality. These alternative measures did not enhance performance (lower AURC is preferable), thereby reinforcing the validity of adopting the mean and variance.
>
> Moreover, it's important to note that KNN, GMM, and density-based methods may entail higher computational costs compared to our approach, given the high efficiency of utilizing mean and variance in the proposed method.
>
> Thank you for your insightful feedback. Since the fundamental mean and variance calculation already yields satisfactory performance in our method, delving into more advanced designs holds promise for further enhancing our method in the future.
>
> **Q3**：The paper only compare failure detection task; the impact of TAL on classification accuracy is not analyzed ( accuracy is noly provided in the table but not discussed).
>
> Thank you for highlighting the importance of conducting a more thorough analysis of the impact of our method. In the revised version of our paper, we have incorporated the discussion on classification accuracy:
> - We have analyzed the impact of TAL on classification accuracy across different datasets.
> - We have discussed the relationship between improved failure detection and classification performance.
> - We have compared these results with baseline methods.
>
> This supplementary analysis will offer a more comprehensive view of the advantages of TAL, demonstrating its value not only for failure detection but also for the overall model performance.
>
> **Q4**：Minor error: "pred" and "GT" in Fig.2 should be swapped.
>
> We have corrected this typo in the revised manuscript.
>
> **Q5**：In Section 3.2 "We empirically find this simple strately works well in our experiments." Please explain why the mean and variance can represent the features of a sample to measure typicality?
>
> We selected the mean/variance based on insights from CORES (CVPR2024), indicating that ID samples exhibit greater magnitudes and variations in responses compared to OOD samples. Fig.  5 (a) visually represents the disparity in mean responses between ID (typical) and OOD (atypical) samples.
>
> Moreover, Fig. 6 (b) depicts the correlation between typicality and density utilizing mean and variance, suggesting that typicality can serve as a substitute for density. Density, calculated by Gaussian KDE (Kernel Density Estimation), represents the likelihood of observing a data point within the distribution.
>
> **Q6**：The confidence calibration methods aim to achieve more reliable confidence estimates, but fail in failure detection task. Could please explain that?
>
> Thank you for raising this important question about the difference between confidence calibration and failure detection. This distinction is crucial for understanding the limitations of traditional confidence calibration methods in failure detection tasks.
>
> Confidence calibration methods are designed to ensure that predicted confidence levels align with actual accuracy rates. For instance, in a perfectly calibrated model, out of 10 samples predicted with a confidence level of 0.9, we would expect 9 to be correct and 1 to be incorrect. While this alignment is crucial for calibration purposes, it may not suffice for effective failure detection.
>
> In failure detection tasks, our objective differs. We aim for high-confidence predictions to be consistently accurate. The occurrence of even one inaccurate prediction among high-confidence samples poses a significant issue. This is due to:
>
> - Failure detection requires the identification of all potential errors, including those within high-confidence predictions.
>
> - In critical applications, it is imperative not to overlook any failures, irrespective of the overall calibration performance.
>
> Our approach, TAL, mitigates this issue by enhancing the reliability of high-confidence predictions, particularly tailored for failure detection tasks. TAL enables us to enhance the detection of potential failures, even in scenarios where conventional calibration metrics indicate the model is well-calibrated.
>
> **Q7**：The OoD data and ID data are presented in equal proportions. How does the ration of them affect the results.
>
> Thank you for raising this concern. To tackle this issue, we have conducted experiments in three settings: new failure detection, out-of-distribution (OOD) detection, and old failure detection.
>
> Our results indicate the following:
> TAL outperforms baseline methods in all three settings;
> For OOD detection, specialized methods perform better than TAL;
> However, most OOD-specific methods exhibit reduced effectiveness in general failure detection tasks.
>
> The robust performance of TAL across all settings underscores its versatility as a solution for various failure detection scenarios. As for the data proportion, we observed that it does not affect our method much. A detailed analysis of these findings will be included in the revised manuscript.

---

> > ### Comment · Reviewer_yqWV · 2024-08-10
> > **Satisfied with the rebuttal and another question.**
> >
> > Thanks for your detailed rebuttal. I appreciate your efforts to explain my questions. Please update the corresponding explanation in the revised paper for better quality.
> >
> > Besides, I have another question: What is the __underlying reason__ for the observed difference in mean values of the features between in-distribution (ID) and out-of-distribution (OoD) data?  Please provide further elaboration on this point.

---

> > > ### Author Response · Authors · 2024-08-11
> > > **Responses to the question**
> > >
> > > **Q1**: What is the underlying reason for the observed difference in mean values of the features between in-distribution (ID) and out-of-distribution (OoD) data? Please provide further elaboration on this point.
> > >
> > >
> > > Thank the reviewer for the question.
> > >
> > >
> > > The underlying reason for the observed difference in mean values of the features between in-distribution (ID) and out-of-distribution (OOD) data, according to CORES [ref 1], is that the convolutional kernels in a trained deep neural network are inherently tuned to extract fundamental attributes of input samples. These kernels exhibit strong responses to patterns they recognize - inputs that are consistent with the training data distribution (ID). In contrast, their response diminishes for patterns they do not recognize, characteristic of out-of-distribution (OOD) inputs.
> > >
> > > - [ref1] Tang, Keke, et al. CORES: Convolutional Response-based Score for Out-of-distribution Detection. CVPR2024.
> > >
> > >
> > >
> > >
> > > If you have any further questions or need additional clarification, please don't hesitate to let us know. We are more than happy to provide more details.

---

> > > > ### Comment · Reviewer_yqWV · 2024-08-12
> > > >
> > > > Thank you for your efforts during the rebuttal period. I think my concerns have been addressed adequately. Considering the questions and suggestions from the other reviewers, and trusting that the authors will include the corresponding experiments, necessary explanations and other recommended components, I decide to raise my score to 7.

---

> > > > > ### Author Response · Authors · 2024-08-13
> > > > > **Thanks to reviewer**
> > > > >
> > > > > We are deeply grateful for your thorough review and positive recognition of our work. The insightful comments demonstrate a deep understanding of our research, which we greatly value. Your initial evaluation has been a significant source of encouragement for our team, motivating us to further refine and improve our paper. We truly appreciate your professionalism and constructive approach throughout the review process.

---

### Official Review · Reviewer_hbz4 · 2024-07-11

**Soundness:** 3
**Presentation:** 3
**Contribution:** 3
**Rating:** 6
**Confidence:** 2

**Summary:**

In this work, the authors propose a new approach to failure detection from DNN predictions. They suggest that data samples can be classified as either typical or atypical. The latter includes ambiguous, out of distribution samples or ill annotated data.
In order to circumvent this limitation, the authors introduce a new training objective which leverages a simple and effective set of features to determine whether a data sample is typical or not during both training and inference. In particular, they propose to store first and second order moment statistics from training samples and measure the Wasserstein distance from new samples to these distributions. This distance is then leveraged in order to weight the cross entropy logits. As a result, a large distance increases the logits magnitude and thus reduces the requirements from the model. This enables the use of the logits direction as a confidence metric.
The authors propose to evaluate the proposed method on Cifar10, Cifar100 and ImageNet for both CNNs ViT, showcasing the added value of the method in these experiments.

**Strengths:**

I am not very familiar with the field, but this work seems original to me and manages to achieve a non-negligible improvement over recent methods while remaining fairly simple.
Furthermore, the proposed method does not appear to be specific to the model architecture but rather to the training loss which is very commonly used: the cross-entropy.
The current presentation of the paper is clear.

**Weaknesses:**

I have two minor concerns regarding this work:
1. in its current form the empirical quantitative evaluation is not fully convincing as it is bounded to computer vision tasks and models with very few comparison points on large scale datasets.
2. the authors insist on the intuition behind the added value with an explicit illustration in the motivation but I did not see an explicit evaluation of the typical vs atypical detection.

**Questions:**

My main question is: could the authors derive a small set of atypical and typical data (labelled set) from a known ood dataset for Imagenet such as [1] and measure the ability of the proposed method to actually separate typical vs atypical data?
i believe this work would benefit from such validation

[1] Galil I, Dabbah M, El-Yaniv R. A framework for benchmarking class-out-of-distribution detection and its application to imagenet. ICLR 2023

**Limitations:**

I believe the authors had a fair description of the current limitations of their work. I would suggest adding a word regarding the specificity of the training loss in the current presentation of the method.

---

> ### Author Rebuttal · Authors · 2024-08-07
>
> Thank you for your valuable comments and kind words to our work.
>
> **Q1**： in its current form the empirical quantitative evaluation is not fully convincing as it is bounded to computer vision tasks and models with very few comparison points on large scale datasets.
>
> Thank you and the results on ImageNet are presented in Tab.  8  (see rebuttal.pdf).
>
> It's worth noting that recent failure detection works, such as Openmix [CVPR2023] and FMFP [ECCV2022], primarily focused on CIFAR, and we followed the same setting for a fair comparison.
> However, the experiments on the large-scale ImageNet further demonstrate the effectiveness of the proposed TAL.
>
> **Q2**： the authors insist on the intuition behind the added value with an explicit illustration in the motivation but I did not see an explicit evaluation of the typical vs atypical detection.
>
> We would like to clarify that, TAL does not explicitly identify typical or atypical samples but dynamically adjusts the training optimization process for different samples by calculating a value that reflects the degree of typicality, in order to alleviate the overconfidence issue.
> In other words, TAL tailors the optimization strategy for typical and atypical samples.
>
> Specifically, for typical samples, our method prioritizes direction optimization by enhancing the TAL loss optimization with a large $\tau$.
>
> **Q3**：Could the authors derive a small set of atypical and typical data (labeled set) from a known ood dataset for Imagenet such as [1] and measure the ability of the proposed method to actually separate typical vs atypical data? i believe this work would benefit from such validation.
>
> We have added a visualization of typical and atypical samples in the revised manuscript (as shown in Fig. 7 (b), which demonstrates the distinction between typical and atypical samples. Thank you for this valuable suggestion, and we believe this illustration may facilitate the understanding.
>
> Conversely, for atypical samples, the small $\tau$ (i.e., large 1 - $\tau$) emphasizes the optimization of the CE loss that considers
>  both direction and magnitude. This may prioritize the magnitude and reduce the impact of atypical samples on the direction, making direction a more reliable confidence indicator.
>
> Thank you and we will add the above elaboration to the revision.

---

> > ### Comment · Reviewer_hbz4 · 2024-08-08
> > **further question**
> >
> > I would like to thank the authors for their response. I am still confused about the typical vs atypical detection.
> > In the response, you state that the proposed method "dynamically adjusts the training optimization process for different samples by calculating a value that reflects the degree of typicality". My question would be, then why can't we derive a typical vs atypical detection method from this value? If we can, it would be interesting to do so in order to empirically validate that the intuition behind the proposed method actually corresponds to what occurs during training.

---

> ### Author Response · Authors · 2024-08-09
> **Responses to the question**
>
> **Q1**： My question would be, then why can't we derive a typical vs atypical detection method from this value? If we can, it would be interesting to do so in order to empirically validate that the intuition behind the proposed method actually corresponds to what occurs during training.
>
> Yes. Our $\tau$ can indeed be used to distinguish between typical and atypical samples, similar to other methods for assessing typicality (such as density). Density is calculated using Gaussian kernel density estimation (Gaussian KDE) and represents the likelihood of observing a particular data point within a distribution. Fig. 6(b) in our rebuttal.pdf provides a scatter plot showing the relationship between density and our calculated typicality with $\tau$, which demonstrates a positive correlation. The advantage of our method is that it consumes fewer resources and computes quickly, making it suitable for use during the training process. Thank the reviewer for the suggestion.

---

### Official Review · Reviewer_9uPn · 2024-07-12

**Soundness:** 2
**Presentation:** 3
**Contribution:** 3
**Rating:** 7
**Confidence:** 4

**Summary:**

This paper proposes a novel training method for improving the failure detection ability of classification models. The authors argue "overconfidence" may be in part due to overfitting of a model to the one hot labels of "atypical" samples. In order to mitigate this issue, a dynamic modification of the LogitNorm training method is proposed where the logit direction is more focused on for more "typical" samples. The proposed TAL is evaluated on image classification failure prediction benchmarks, showing promising performance on CIFAR data.

**Strengths:**

- I believe the problem setting of failure detection (Fig. 2) is understudied and it is important for more work to focus on it.
- The results on CIFAR data are promising - TAL is able to perform well compared to a number of existing training-based approaches, both at detecting just misclassifications and mixed OOD + misclassifications.
- The high-level idea/motivation of TAL, focusing on the difference between "atypical" samples, i.e. those on the tail of the training distribution, and optimising them differently is intuitive and appealing.

**Weaknesses:**

As the reviewing burden has been heavy for this conference (6 papers) please understand that I can only dedicate so much time to this paper. Thus, I may have made mistakes in my understanding of the paper, and I welcome the authors to correct me if this is the case.

1. Missing comparisons. A number of methods that should be compared against are missing. CRL [1] (similar to TAL) is a learning-based approach for better misclassification detection, whilst SIRC [2] is a post-training method explicitly designed for rejecting both ID errors and OOD samples. It would also be better to clearly distinguish training and post-training approaches in the results.
2. No risk-coverage curves presented. Although scalar scores show overall failure detection performance, plotting out RC curves in my opinion tells a lot more about the performance of an approach.
3. Odd ImageNet results. The accuracy for ResNet-50 on this benchmark is far below the standard accuracy achievable using cross entropy training (~76%) and standard augmentations. Thus I am yet unconvinced that this approach is able to scale up to more realistic data. CIFAR is a rather special optimisation scenario where training accuracy converges to 100%. Models are typically extremely overparameterised and the data are much smaller scale than real-world CV applications. Thus, CIFAR results may not generalise to real applications.
4. Missing references/attribution to existing work. There are a number of other published work that targets new failure detection that are not cited [2,3,4]. In particular [2] already discusses the behaviour described in Appendix B at length.
5. Complexity of method. The actual method of TAL is quite complex, with a number of stages (d calculation, \tau calculation), hyperparameters (Tmin/max, queue length) and ad hoc design choices (mean/variance calced along feature dimension, wasserstein distance, using \tau to mix TAL and CE). This complexity limits ease of adoption, and makes it difficult to understand the reasons for TAL's efficacy. A more thorough ablation would greatly improve the understanding of the importance of each choice, aiding potential practitioners. Similarly, some intermediate experiments, (e.g. showing how error-rate changes according to "typicalness") would also make the method of TAL more convincing.
6. Missing details. It is unclear to me what confidence score is used in the end for TAL. Moreover, how is \tau set during inference? Is it fixed or is it calculated in the same way as during training? What is the OOD dataset for ImageNet?


References

[1] Moon et al., Confidence-Aware Learning for Deep Neural Networks

[2] Xia and Bouganis, Augmenting Softmax Information for Selective Classification with Out-of-Distribution Data

[3] Narasimhan et al., Plugin estimators for selective classification with out-of-distribution detection

[4] Kim et al., A Unified Benchmark for the Unknown Detection Capability of Deep Neural Networks

**Questions:**

1. As CRL requires training, but SIRC doesn't, and given the length of the rebuttal period, I would be happy with a comparison with just SIRC.
2. I'd like to see a couple of risk-coverage curves to get a clearer idea of how TAL impacts failure detection (of just TAL vs MSP/CE).
3. I would **require** ImageNet results with standard accuracies (~76%) which should be achievable using the recipe from the original resnet paper.
4. Include these references.
5. I would like to see some more ablations if possible (e.g. using a different measure of typicality, using static mixing of CE/TAL. A toy experiment (e..g 2D classification) showing how OOD samples/samples in lower density areas of the training distribution are assigned lower typicality would help a lot as well. An experiment such as finding atypical samples in the test set (using a model's predictive uncertainty) and then showing how those labels are overfitted when the test set is folded into the training set under CE but not under TAL would also improve the paper in my opinion.
6. Please clarify this.

Depending on the number of questions/weaknesses addressed I will be happy to raise my score up to 6.

Some additional food for thought for the authors. Knowledge Distillation may also help in softening supervision on atypical samples. Perhaps it could be incorporated into future work. I also think it would be good to clarify that the authors' use of "typical" is different to that commonly found in information theory.


As the authors have successfully rebutted my queries, as well as considering the relative scarcity of effective approaches for the problem setting (compared to e.g. OOD detection), I have decided to raise my score to 7.

**Limitations:**

See above

---

> ### Author Rebuttal · Authors · 2024-08-07
>
> Thank you for your valuable feedback on our paper.
>
> **Q1**: Missing comparisons. A number of methods that should be compared against are missing. As CRL requires training, but SIRC doesn't, and given the length of the rebuttal period, I would be happy with a comparison with just SIRC.
>
> The comparison results with SIRC are shown in Tab. 8 (see rebuttal.pdf),  and we will add them to the final version. However, we are deeply sorry that we are unable to finish the results of CRL due to the limited time and resources during the rebuttal period.
>
> **Q2**：No risk-coverage curves presented. Although scalar scores show overall failure detection performance, plotting out RC curves in my opinion tells a lot more about the performance of an approach.
>
> The risk-coverage curves are shown in Fig. 7(a), and we will them to the main paper.
>
> **Q3**：Odd ImageNet results. The accuracy for ResNet-50 on this benchmark is far below the standard accuracy achievable using cross entropy training (~76\%) and standard augmentations. Thus I am yet unconvinced that this approach is able to scale up to more realistic data. CIFAR results may not generalise to real applications.
>
> We apologize for the subpar ImageNet results in our initial submission. Due to time constraints, our focus was primarily on conducting comprehensive experiments on CIFAR, leading to insufficient training time allocated for ImageNet and subsequent underperformance.
>
> To rectify this issue, we have conducted thorough experiments on ImageNet. The revised results are detailed in Tab. 8, showcasing the efficacy of our approach in scaling effectively to larger and more realistic datasets like ImageNet.
>
> **Q4**：Missing references/attribution to existing work. There are a number of other published work that targets new failure detection that are not cited [2,3,4].
>
> Thank you for your reminder. We have added these references in the revised manuscript.
>
> **Q5**：Complexity of method. The actual method of TAL is quite complex, with a number of stages (d calculation, $\tau$ calculation), hyperparameters (Tmin/max, queue length) and ad hoc design choices (mean/variance calced along feature dimension, wasserstein distance, using $\tau$ to mix TAL and CE). This complexity limits ease of adoption, and makes it difficult to understand the reasons for TAL's efficacy. A more thorough ablation would greatly improve the understanding of the importance of each choice, aiding potential practitioners. Similarly, some intermediate experiments, (e.g. showing how error-rate changes according to "typicalness") would also make the method of TAL more convincing.
>
> The following provides additional explanations and experiments to address your concerns:
>
> - 1. Hyperparameters and Intermediate Variables:
> The hyper-parameters include queue length, Tmin, and Tmax, with ablation studies provided. The intermediate variable d is normalized within each batch to derive $\tau$, capturing the relative typicality during training.
>
> - 2. Feature Representation:
> We selected the mean/variance based on insights from CORES (CVPR2024), indicating that ID samples exhibit greater magnitudes and variations in responses compared to OOD samples. Fig.  5 (a) visually represents the disparity in mean responses between ID (typical) and OOD (atypical) samples.
>
> - 3. Ablation of Typicality Measures:
> As depicted in Fig. 5 (b), we have conduced extra ablation experiments with K-nearest neighbor (KNN) distance and Gaussian Mixture Models (GMM) to assess typicality. These alternative measures did not enhance performance (lower AURC is preferable), thereby reinforcing the validity of our selection of mean/variance criteria.
>
> - 4. Reasons for TAL's effectiveness: Our approach tailors the optimization strategy for typical and atypical samples to alleviate overconfidence.
> In particular, for typical samples, our method prioritizes direction optimization by enhancing the TAL loss optimization with a large $\tau$.
> Conversely, for atypical samples, the small $\tau$ (i.e., large 1 - $\tau$) emphasizes the optimization of the CE loss that considers
>  both direction and magnitude. This may prioritize the magnitude and reduce the impact of atypical samples on the direction, making direction a more reliable confidence indicator.
>
>
> - 5. Error Rate vs. typicality:
> We have included an experiment illustrating the variation in error rates with typically (Fig.  6 (a)). This experiment offers valuable insights into the behavior of TAL.
>
> Thank you and we will add the elaboration to the revision.
>
> **Q6**：Missing details. It is unclear to me what confidence score is used in the end for TAL. Moreover, how is $\tau$ set during inference? Is it fixed or is it calculated in the same way as during training? What is the OOD dataset for ImageNet?
>
> - Confidence Score and Inference:
> In the inference phase, TAL functions akin to a model trained with conventional cross-entropy. We calculate the cosine value of the predicted direction to derive the confidence score. This aligns with our focus on direction as a more dependable confidence metric, as highlighted in the introduction and the response to Q5-4 (Reasons for TAL's effectiveness).
>
> - Calculation of $\tau$: $\tau$ is not used during inference, but only used during training for adjusting the magnitude/direction optimization according to the sample typicalness, as mentioned the response to Q5-4.
>
> - OOD Dataset for ImageNet. We used the Textures dataset as the OOD dataset for ImageNet experiments.
>
> Thank you and we will add the above clarifications to the final version.

---

> > ### Comment · Reviewer_9uPn · 2024-08-08
> >
> > Thank you for taking the time and effort to address my questions (which I think have strengthened the paper). Although I am broadly happy with the authors' response, I still have a few things I want to clarify.
> > - Which version of SIRC did you use? SIRC is just a method to combine scores and it's unclear which scores are combined. In the original paper the best-performing combination was Entropy+Residual for example. SIRC is also typically used with softmax scores, so when applied to TAL, is it used with the cosine measure or a softmax score?
> > - Could you include the training recipe for the updated ImageNet results? Also, how are the OOD/ID datasets balanced i.e. what is the ratio between the number of incorrect ID predictions and OOD samples?
> > - The use of cosine distance as a confidence score is important to the story, and helps me understand the approach a lot better. I would suggest making this clear to the reader early on (e.g. in Figure 1). I would generally suggest editing the manuscript to provide a clearer story, i.e. incorporating some of the new plots into the method section. Hopefully the authors see the value in this.
> > - How are the error/density plots calculated? On the training set at a particular snapshot? On the test set using the final queue?
> > - FMFP appears to fail on ImageNet. I am not familiar with the method but it would be good to offer insight here, as well as caution that the method may suffer from poor scalability.

---

> ### Author Response · Authors · 2024-08-09
> **Responses to the questions**
>
> We thank the reviewer for valuable suggestions and insights, which have significantly improved our manuscript.
>
> **Q1**: Which version of SIRC did you use? SIRC is just a method to combine scores and it's unclear which scores are combined. In the original paper, the best-performing combination was Entropy+Residual for example. SIRC is also typically used with softmax scores, so when applied to TAL, is it used with the cosine measure or a softmax score?
>
> In our experiments, we utilized the most effective variant of SIRC for the FD task, i.e. SIRC_MSP_z. This configuration combines Maximum Softmax Probability (MSP) with the feature. For the TAL+SIRC combination, we opted for a cosine measure paired with the feature.
>
> **Q2**: Could you include the training recipe for the updated ImageNet results? Also, how are the OOD/ID datasets balanced i.e. what is the ratio between the number of incorrect ID predictions and OOD samples?
> The use of cosine distance as a confidence score is important to the story, and helps me understand the approach a lot better. I would suggest making this clear to the reader early on (e.g. in Figure 1). I would generally suggest editing the manuscript to provide a clearer story, i.e. incorporating some of the new plots into the method section. Hopefully the authors see the value in this.
>
> - Training protocol for the updated ImageNet experiments: We employed a ResNet50 architecture as the base model.
>     For data preprocessing, we implemented standard augmentation techniques. These include random cropping, horizontal flipping, and normalization.
> Our training configuration includes a batch size of 256 and an initial learning rate of 0.1. The learning rate was managed by a StepLR scheduler, which reduced it by a factor of 0.1 every 30 epochs.
> We chose SGD as our optimizer, with a momentum of 0.9 and weight decay of 0.0001. The total training duration was set to 90 epochs.
> To accelerate training, we utilized distributed training across four 3090 GPUs.
> For the TAL-specific parameters, we kept Tmax and Tmin unchanged. The queue length and its initial epochs were adjusted proportionally, following the same ratio as in CIFAR experiments. This approach ensures consistency across different datasets while accounting for the larger scale of ImageNet.
>
> - Ratio of OOD/ID data: ID (correct+incorrect) : OOD = 1:1 . For baseline，incorrect ID: OOD = 451:1880 ;  For TAL, incorrect ID: OOD = 445:1880.
>
> -  Thanks for the suggestion. In the new revision, we will revise Figure 1 for a more clear explanation of our method. We will explicitly illustrate the training and testing processes respectively. We believe this addition will effectively address the reviewer's question and improve the clarity of our paper.
>
> **Q3**： How are the error/density plots calculated? On the training set at a particular snapshot? On the test set using the final queue?
>
> Density, calculated using Gaussian KDE (Kernel Density Estimation), represents the likelihood of observing a data point within the distribution. The density is computed using the ImageNet test set. Typicality, denoted by $\tau$ in the original paper, is determined by calculating the distance of each test sample to the training samples, using a historical queue constructed from the mean and variance of the training set (recomputed by the last snapshot) to obtain typically.
>
> **Q4**：
> FMFP appears to fail on ImageNet. I am not familiar with the method but it would be good to offer insight here, as well as caution that the method may suffer from poor scalability.
>
> FMFP employs Stochastic Weight Averaging (SWA) and Sharpness-Aware Minimization (SAM). SWA and SAM can suffer from sensitive hyperparameters for different datasets. Since FMFP did not report experimental results on ImageNet, we directly transferred the hyperparameters from CIFAR to ImageNet. We speculate that FMFP might need to carefully adjust the hyperparameters on ImageNet to achieve sufficient accuracy. On the other hand, for our TAL method, we also directly transfer the hyperparameters from CIFAR to ImageNet and achieve significant improvements over baselines (shown in Table 8), which demonstrates the stability of our algorithm.

---

> > ### Comment · Reviewer_9uPn · 2024-08-09
> >
> > Thanks for the further clarification.
> >
> > One final thing, I’d still like to see the results for SIRC with the residual score. Although the residual score is somewhat unreliable depending on the distribution shift, it seems to do well in OOD detection here, so I would expect it to still help a fair amount. I am surprised in this case that you state the version with the feature norm is better.

---

> ### Author Response · Authors · 2024-08-09
> **Responses to the question**
>
> Table 1: The performance differences in Old FD task among various SIRC variants.
> | Method | AUROC$\uparrow$ | FPR95$\downarrow$ | AURC$\downarrow$ |
> |--------|-------|-------|------|
> | SIRC_H_z | 85.41 | 67.62 | 74.80 |
> | SIRC_MSP_z | 86.11 | 63.67 | 72.91 |
> | SIRC_doctor_z | 86.12 | 64.76 | 72.88 |
> | SIRC_H_res | 85.54 | 66.67 | 74.44 |
> | SIRC_MSP_res | 86.18 | 63.40 | 72.70 |
> | SIRC_doctor_res | 86.16 | 65.71 | 72.73 |
> |TAL | 87.11 | 64.93|64.66|
> |SIRC_cos_z(TAL)|87.15|63.66|64.55|
>
> The Table shows the performance for all of the variants of the SIRC method on ImageNet.
> We observe that SIRC_MSP_z and SIRC_MSP_res achieve similar performance. Our TAL algorithm can significantly surpass the baselines. Thank the reviewer for the question.

---

> > ### Comment · Reviewer_9uPn · 2024-08-10
> >
> > Thanks for the response and apologies for the delayed reply.
> >
> > In my earlier comment, I was referring to the performance of SIRC on failure detection that includes OOD data. SIRC is not meant to improve detection performance when there isn't semantically shifted data. So to be more precise, I would like to see the results, on ImageNet+Textures, of SIRC with the residual score.
> >
> > I'd like to clarify that SIRC is a post-hoc method and as such doesn't directly compete with TAL (and as you have demonstrated can complement TAL). As such I am interested in if the best version of SIRC is able to further boost TAL's performance.
> >
> > I would also suggest that the authors revise the tables in their paper to more clearly delineate training-free and training-based approaches.
> >
> > Finally, I have been pleasantly surprised with this rebuttal period and feel like the quality of the manuscript will be significantly improved as a result. If the authors can address my final requests I will increase my score to 7.

---

> > > ### Author Response · Authors · 2024-08-11
> > > **Responses to the question**
> > >
> > > **Q1**:  So to be more precise, I would like to see the results, on ImageNet+Textures, of SIRC with the residual score.
> > >
> > > We greatly appreciate the reviewer's insightful suggestion. Following your recommendation, we combined our TAL method with different versions of SIRC and evaluated their performance on ImageNet+Textures. The results are presented in the table below.
> > >
> > > | Method | Old setting FD | | | OOD Detection | | | New setting FD | | | ID-ACC |
> > > |--------|----------------|-------|-------|---------------|-------|-------|----------------|-------|-------|--------|
> > > | | AURC↓ | FPR95↓ | AUROC↑ | AURC↓ | FPR95↓ | AUROC↑ | AURC↓ | FPR95↓ | AUROC↑ | |
> > > | TAL | 64.66 | 64.93 | 87.11 | 290.5 | 47.66 | 87.51 | 338.45 | 50.11 | 88.29 | 76.43 |
> > > | TAL+SIRC_cos_z | 64.55 | 63.66 | 87.15 | 288.23 | 46.91 | 87.88 | 336.56 | 49.68 | 88.35 | 76.43 |
> > > | TAL+SIRC_cos_res | 65.74 | 66.62 | 86.69 | 283.77 | 45.42 | 88.50 | 333.16 | 49.20 | 88.47 | 76.43 |
> > >
> > > The results reveal that TAL combined with SIRC and residual scores demonstrates superior performance in OOD-D and New FD tasks.
> > > On the task of Old FD task without OOD data during inference, TAL combined with SIRC and features achieved the best results.

---

> > > > ### Comment · Reviewer_9uPn · 2024-08-12
> > > >
> > > > Thanks for the results. I will be also expecting non-TAL SIRC Res results across all ImageNet cols in the final manuscript.
> > > > Accordingly I will raise my score to 7.
> > > >
> > > > It's interesting that SIRC leads to more degradation on old FD with the cosine score. Maybe this is because it is not a softmax score and so the values are generally farther away from 1.0 and closer to  (see Fig. 4 of the SIRC paper). I would be good to see some more analysis *if* the authors have time. I have a hunch that taking the square or cube root of the cosine score before SIRC would reduce degradation in old FD.
> > > >
> > > > Thank you for a fruitful rebuttal period.

---

> > > > > ### Author Response · Authors · 2024-08-13
> > > > > **Responses to the question and Thanks to reviewer**
> > > > >
> > > > > **Q1**: It's interesting that SIRC leads to more degradation on old FD with the cosine score. Maybe this is because it is not a softmax score and so the values are generally farther away from 1.0 and closer to (see Fig. 4 of the SIRC paper). I would be good to see some more analysis if the authors have time.
> > > > >
> > > > > Thank the reviewer for the suggestion. More results are presented in the table below.
> > > > >
> > > > > | Method | Old setting FD | | | OOD Detection | | | New setting FD | | | ID-ACC |
> > > > > |--------|----------------|-------|-------|---------------|-------|-------|----------------|-------|-------|--------|
> > > > > | | AURC↓ | FPR95↓ | AUROC↑ | AURC↓ | FPR95↓ | AUROC↑ | AURC↓ | FPR95↓ | AUROC↑ | |
> > > > > |MSP_res(SIRC) | 72.70 | 63.40 | 86.18 | 276.45 |  30.85 | 90.73 | 330.73 | 37.27 | 89.79 | 76.13 |
> > > > > | TAL | 64.66 | 64.93 | 87.11 | 290.5 | 47.66 | 87.51 | 338.45 | 50.11 | 88.29 | 76.43 |
> > > > > cos_z(TAL) | 64.55 | 63.66 | 87.15 | 288.23 | 46.91 | 87.88 | 336.56 | 49.68 | 88.35 | 76.43 |
> > > > > | cos_res(TAL) | 65.74 | 66.62 | 86.69 | 283.77 | 45.42 | 88.50 | 333.16 | 49.20 | 88.47 | 76.43 |
> > > > > | (cube_root cos)_res(TAL) | 65.62 | 65.91 | 86.73 | 284.09 | 44.94 | 88.44 | 333.38 | 48.51 | 88.43 | 76.43 |
> > > > > | (square_root cos)_res(TAL) | 65.58 | 66.05 | 86.75 | 284.19 | 45.10 | 88.42 | 333.45 | 48.68 | 88.42 | 76.43 |
> > > > >
> > > > > For completeness of our work, we will include non-TAL SIRC Res results across all ImageNet cols in the final manuscript.
> > > > >
> > > > > Additionally, we implemented the reviewer's suggestion to modify the range of cosine similarity. However, the square or cube root of the cosine score did not significantly alter the experimental results. We will add an explicit discussion to explore the problem in the new revision.
> > > > >
> > > > > We are deeply grateful for the reviewer's exceptional feedback. The reviewer's dedication contributes significantly to the quality improvements of our paper.
> > > > > The insightful comments and suggestions demonstrate the reviewer's profound expertise. We sincerely appreciate the time and effort invested in providing such comprehensive and constructive comments.

---

### Official Review · Reviewer_RLx5 · 2024-07-27

**Soundness:** 2
**Presentation:** 3
**Contribution:** 2
**Rating:** 6
**Confidence:** 4

**Summary:**

The paper proposed a method for detecting incorrect prediction (Failure Detection). The main hypothesis behind the method is that cross-entropy either increases the logit magnitude or aligns the logit direction to align it with the ground label, which can cause discrepancy on atypical samples at test time. Previous methods, such as LogitNorm, only focussed on the logit magnitude; the proposed method tries to adaptively align training samples based on a metric that measures the typicalness of each sample. The paper also proposes a new setting, 'New FD', for evaluating failure detection methods. Extensive experiments were conducted on CIFAR-10/100, with some preliminary results on ImageNet (not benchmarked properly).

**Strengths:**

1. The paper is well-written and easy to follow; the proposed method is intuitive and model-agnostic.

2. Extensive experiments on CIFAR-10/100 show that the proposed method is effective for Failure Detection compared to the existing baselines.

3. The new proposed setting of New-FD is interesting, providing a unified view of both OOD shift and covariate shift.

4. An ablation study was conducted to show how hyper-parameters for the experiments were chosen.

**Weaknesses:**

While the proposed method looks promising, there are a few weaknesses and areas where the manuscript can be further improved:

1. Most of the experiments were conducted on the CIFAR dataset (Table 1). Results on ImageNet are inconclusive; only MSP was used as a baseline for comparison. It would be helpful to add more baselines for comparison.


2. Experiments on ViT were conducted on CIFAR. It is not clear why CIFAR was chosen as a dataset for ViT instead of ImageNet.

3. As SSL-based models or fine-tuning pre-trained foundational models are becoming state-of-the-art models for various tasks, it is unclear if the proposed method can be applied in that setting. Experiments should be conducted on fine-tuning pre-trained foundational models.

4. Some parts of section 3 are not clear:

      a. How are mean and variance calculated for each sample feature?
      b. Some samples can be incorrectly classified in some parts of the training; how do you handle such samples to update HFQ?
      c. How is $d_{min}$ and $d_{max}$ are calculated?
      d. Variables are not explained, e.g., equation 3.

5. It is not clear how equation 10 works. $\tau$ measures 'typicalness'; a value of 1 indicates a typical sample, and 0 indicates an atypical sample. The CE loss is weighted by $(1-\tau)$, i.e., the atypical sample ($\tau=0$) will only be trained by CE loss, which is counterintuitive.

6. Authors only use SVHN for evaluating OOD shifts; systematic evaluation on the WILDS dataset [1] should be added.

7. The need for the New-FD setting (detecting both is not explained, and the authors chose to evaluate the baselines in this setting. The experimental setup (New-FD) is unfair for baselines proposed to detect only OOD shifts (semantic shift). The proposed method should be evaluated for semantic and covariate shifts independently.

[1] Koh et al., WILDS: A Benchmark of in-the-Wild Distribution Shifts

**Questions:**

1. In Sec 4.2, which model did you use? It is not clear from the manuscript.

2. What are dynamic magnitudes?  Why do we need it?

3. > In this manner, for atypical samples, a higher value
of $T(\tau)$ reduces the influence that pulls them towards the label direction"

Can you explain more how does T influence the direction?

4. Explain more about equation 10. Why atypical samples optimized only by CE loss?

**Limitations:**

Yes

---

> ### Author Rebuttal · Authors · 2024-08-07
>
> We thank the reviewer for the valuable comments and suggestions.
>
> **Q1**: Most experiments were conducted on CIFAR. Results on ImageNet are inconclusive; only MSP was used as a baseline for comparison.
>
> Thank you for the reviewers' comments. The question about ImageNet is a common concern among the reviewers. Due to character limits, the response to this important question can be found in the main rebuttal.
>
> **Q2**: Experiments on ViT were conducted on CIFAR. It is not clear why CIFAR was chosen as a dataset for ViT instead of ImageNet.
>
>  We would like to humbly clarify that, previous works mainly conducted experiments on CIFAR and they did not report the results on Imagenet. Therefore, we adopted CIFAR for ViT for a fair comparison across all baselines. The experiments on ViT were made to demonstrate the model-agnostic nature of our method, showing its compatibility with both CNN and ViT architectures.
>
> **Q3**：As SSL and fine-tuning pre-trained models become state-of-the-art, it is unclear if TAL method applies.
>
> We conducted experiments on fine-tuning pre-trained models by freezing the feature extractor and only training the classifier. However, the results were unsatisfactory. We hypothesize this is because our method essentially avoids overfitting to atypical features. The frozen feature extractor limits the effectiveness of our method, as only the classifier layer doesn't have enough model capacity.
>
> In the discussion period, we will continue exploring fine-tuning all layers of the network to see if our method can still be effective. Going further, we will also explore combining our method with other OOD detection algorithms.
>
> **Q4**：Some parts are not clear:
> a. How are mean and variance calculated for features? b. How do you handle incorrect samples to update HFQ? c. How is $\tau$ and `d' are calculated? d. Variables are not explained, e.g., equation 3.
> Thank you for your detailed feedback. We will clarify each point in our revised version:
>
> a. We calculate the mean and variance of each sample's feature channels based on insights from CORES (CVPR 2024). This approach stems from the observation that in-distribution samples show larger magnitudes (mean) and variations (variance) in convolutional responses across channels compared to OoD samples, which are a type of atypical sample. As shown in Fig. 5(a), The mean response of OOD samples is smaller than correct ID samples.
>
> b. As stated in line 167 of the main paper, We update the Historical Feature Queue (HFQ) using a first-in-first-out method for correctly predicted samples. We discard the statistics of incorrectly predicted samples.
>
> c.
> We will incorporate the following clarification to Eq. (7) in the revised version.
> For each new batch of samples, we calculate a distance `d' for each sample. We then normalize these distances within the batch (dmin and dmax represent the minimum and maximum distances in the batch). This normalization is crucial as typicalness is relative, and we use it to control the strength of optimization direction.
>
> d. We will provide clearer explanations for all variables, including those in Eq. 3, in our revised version.
>
> **Q5**：It is not clear how equation 10 works. The atypical sample $(1-\tau)$ will only be trained by CE loss, which is counterintuitive.
>
> Our approach tailors the optimization strategy for typical and atypical samples to alleviate overconfidence.
>
> In particular, for typical samples, we prioritize direction optimization by enhancing the TAL loss optimization with a large $\tau$.
>
> Conversely, for atypical samples, the small $\tau$ (i.e., large 1 - $\tau$) emphasizes the optimization of the CE loss that considers
>  both direction and magnitude. This may prioritize the magnitude and reduce the impact of atypical samples on the direction, making direction a more reliable confidence indicator.
>
> Thank you and we will add the elaboration to the revision.
>
> **Q6**：only use SVHN for OOD shifts;  WILDS dataset [1] should be added.
>
> As suggested by the reviewer, we have conducted additional experiments using the WILDS dataset. These results will be included in our revised paper. Due to the limited time, the result table will come in the discussion period.
>
> **Q7**: Evaluation for semantic and covariate shifts independently.
>
> The New-FD setting was introduced in an ICLR 2023 paper, and we incorporated it into our related work section to offer insight into recent advancements in the field. The experiments regarding semantic (OOD) and covariate shifts(Old FD) independently with the evaluation of New FD are shown in Table 8 of rebuttal.pdf.
>
> **Q8**：
> 1.In Sec 4.2, which model did you use?
> 2.What are dynamic magnitudes? Why ?
> 3.how does T influence the direction?
> 4.Explain equation 10.
>
> 1. Model:
> We used ResNet50, and the training details were in the supplementary materials.
>
> 2. Dynamic Magnitudes:
> Dynamic magnitudes regulate the intensity of direction optimization. In contrast to LogitNorm, which employs a fixed logits vector magnitude, our approach adjusts T (magnitude) to modulate the loss function, as depicted in Equation 4. This enables us to dynamically control the optimization strength based on the typicalness of the sample.
>
> 3. Influence of T:
> T controls optimization strength to mitigate overconfidence. For instance, in a binary classification scenario with the logit direction [√3/2, 1/2], raising T causes the sigmoid output approaching to 1, thereby decreasing the cross-entropy (CE) loss. Consequently, higher T values lead to a less intense optimization of the logit direction.
>
> 4. Explain Equation 10:
> As mentioned in our earlier response to Q5, emphasizing the CE loss (not only relying on the CE loss, as it is regulated by $\tau$) for atypical samples enables the optimization of both direction and magnitude. This may help reduce the adverse effects of atypical samples on the direction, enhancing the reliability of direction as a confidence indicator.

---

> > ### Author Response · Authors · 2024-08-08
> > **Experiments on WILDS dataset**
> >
> > **Q6**：only use SVHN for OOD shifts;  WILDS dataset [1] should be added.
> >
> > Here are the failure detection evaluation results of TAL on the WILDS dataset. Thank you and we will supplement these results to the final version.
> >
> > | **Architecture** | **Method** | **AURC↓ (Old FD)** | **FPR95↓ (Old FD)** | **AUROC↑ (Old FD)** | **AURC↓ (OOD Detection)** | **FPR95↓ (OOD Detection)** | **AUROC↑ (OOD Detection)** | **AURC↓ (New FD)** | **FPR95↓ (New FD)** | **AUROC↑ (New FD)** | **ID-ACC** |
> > |------------------|------------|--------------------|---------------------|---------------------|---------------------------|----------------------------|----------------------------|--------------------|---------------------|---------------------|------------|
> > |                  |            | Imagenet vs WILDS                                                                                                                   |
> > | **ResNet50**     | MSP        | 72.73              | 63.95               | 86.18               | 272.93                    | 59.27                      | 87.72                      | 326.85             | 60.19               | 87.42               | 76.13      |
> > |                  | Cosine     | 102.98             | 69.93               | 79.49               | 255.91                    | 68.67                      | 89.85                      | 326.97             | 68.92               | 87.81               | 76.13      |
> > |                  | Energy     | 118.66             | 76.33               | 75.81               | 235.88                    | 37.67                      | 94.22                      | 318.89             | 45.27               | 90.60               | 76.13      |
> > |                  | MaxLogit   | 113.35             | 72.11               | 77.29               | 237.28                    | 38.93                      | 93.97                      | 317.46             | 45.46               | 90.69               | 76.13      |
> > |                  | Entropy    | 74.61              | 67.07               | 85.48               | 259.01                    | 51.20                      | 90.44                      | 316.26             | 54.32               | 89.47               | 76.13      |
> > |                  | mahalanobis| 208.22             | 96.19               | 54.23               | 264.11                    | 77.17                      | 88.20                      | 382.46             | 80.91               | 81.51               | 76.13      |
> > |                  | Residual   | 238.18             | 97.01               | 49.00               | 282.46                    | 81.30                      | 85.09                      | 409.89             | 84.39               | 77.99               | 76.13      |
> > |                  | gradnorm   | 206.99             | 89.66               | 57.88               | 237.31                    | 25.37                      | 94.84                      | 363.03             | 38.02               | 87.56               | 76.13      |
> > |                  | SIRC       | 72.91              | 63.67               | 86.11               | 267.33                    | 52.17                      | 89.03                      | 322.41             | 54.43               | 88.46               | 76.13      |
> > |                  | LogitNorm  | -                  | -                   | -                   | -                         | -                          | -                          | -                  | -                   | -                   | -      |
> > |                  | openmix    | -                  | -                   | -                   | -                         | -                          | -                          | -                  | -                   | -                   | -      |
> > |                  | FMFP       | -                  | -                   | -                   | -                         | -                          | -                          | -                  | -                   | -                   | 60.11      |
> > |                  | TAL (ours)        | 64.66              | 64.93               | 87.11               | 232.11                    | 40.97                      | 94.28                      | 288.67             | 45.55               | 92.91               | 76.43      |

---

> > > ### Comment · Reviewer_9uPn · 2024-08-12
> > >
> > > I'm a little confused by the evaluation using WILDS in this case. My understanding (although I'm not intimately familiar with the dataset) is that it contains various training data and covariate shifted test data. Using it as OOD data to ImageNet doesn't particularly make sense as there's no guarantee with regards to class overlap.
> > >
> > > However, I would encourage reviewer RLx5 to overlook this as the authors have improved their evaluation with the inclusion of ImageNet-1k results

---

> > > ### Comment · Reviewer_RLx5 · 2024-08-12
> > > **Reply to the authors**
> > >
> > > Thank you for providing detailed answers to the questions raised during the review period. The authors have resolved most of the concerns raised during the review period. New experiments on ImageNet and WILDS are really helpful. Since most of my concerns have been resolved, I have increased the score.

---

> > > > ### Author Response · Authors · 2024-08-13
> > > > **Thanks to reviewer**
> > > >
> > > > We sincerely appreciate the time and effort you've invested in our work. Your insightful questions have highlighted areas in our paper that require clearer articulation.
> > > > With your professional suggestions, the quality of our paper is significantly enhanced.
> > > > Thank the reviewer for the careful and thorough reading of our manuscript.

---

### Author Rebuttal · Authors · 2024-08-07

Thanks to all reviewers and ACs for the valuable comments and suggestions. We appreciate that reviewers described our work as `` well-written, intuitive, effective". We are grateful for the reviewers' positive evaluation of our work as ``well-written, intuitive, effective". Each reviewer's feedback has been carefully addressed individually. The manuscript has been revised in accordance with the suggestions provided. Here is a summary of the main concerns raised by the reviewers:


**Comparison**: Most experiments were conducted on CIFAR. Results on ImageNet have added;

- As suggested by the Reviewer, we have expanded our evaluation on ImageNet to include additional baselines, as shown in Table 8 of
 rebuttal.pdf. The evaluation is independently performed on the three settings. The results demonstrate that our approach consistently outperforms existing baselines on ImageNet, aligning with our findings on CIFAR.

- Please note that, because other methods in the community did not report the performance on ImageNet, for a fair comparison, all experiments conducted on ImageNet utilize the same hyper-parameter settings as those used on CIFAR. It is worth mentioning that models with LogitNorm(ICML2022), Openmix(CVPR2023) cannot decently converge on ImageNet, and FMFP(ECCV2022) only achieved 60\% accuracy. Differently, the introduced TAL consistently exhibits enhancements, showcasing its resilience across various data domains.


**Why is TAL effective**: why use mean and variance to represent a sample for measuring typicality?
 We added several visualizations to explain our method's effectiveness (see rebuttal.pdf), including:

- Mean of Features Comparison: ID samples have a higher mean value compared to OOD samples (a type of atypical sample).

- Density and Typicality Relationship: Higher typicality corresponds to lower density, but typicality is faster and more resource-saving.

- Error-over-Typicality Curve: Samples with low typicality show more errors.

- Risk-Coverage Curves.

- Ablation of Other Typicality Measures.


We sincerely appreciate the valuable suggestions and insightful comments given by the reviewers. The authors look forward to further discussions and are willing to address any of your concerns.

Thanks again.

Best regards,

Authors of paper 10758

---

### Decision · Program_Chairs · 2024-09-25

**Decision:**

Accept (poster)

**Comment:**

There was consensus from the reviewers on the work being well-motivated, well-written and the methodology being intuitive. Much of the reviewer's concerns pre-rebuttal centred on the ImageNet results, the small number of such results, and the poor results. Other concerns included missing references/comparisons with existing work. However, significant work by both the authors and reviewers during the rebuttal period discussion appears to have addressed these concerns for the reviewers, in particular with much improved ImageNet results.